# Polar Sparsity: High Throughput Batched LLM Inferencing with Scalable Contextual Sparsity

**Susav Shrestha**
Texas A&M University
sls7161@tamu.edu

**Brad Settlemyer**
NVIDIA
bsettlemyer@nvidia.com

**Nikoli Dryden**
Lawrence Livermore National Laboratory
dryden1@llnl.gov

**Narasimha Reddy**
Texas A&M University
reddy@tamu.edu

## Abstract

Accelerating large language model (LLM) inference is critical for real-world deployments requiring high throughput and low latency. Contextual sparsity, where each token dynamically activates only a small subset of the model parameters, shows promise but does not scale to large batch sizes due to union of active neurons quickly approaching dense computation. We introduce Polar Sparsity, highlighting a key shift in sparsity importance from MLP to Attention layers as we scale batch size and sequence length. While MLP layers become more compute-efficient under batching, their sparsity vanishes. In contrast, attention becomes increasingly more expensive at scale, while their head sparsity remains stable and batch-invariant. We develop Selective Head Attention with hardware-efficient, sparsity-aware GPU kernels, delivering up to $2.2\times$ end-to-end speedups for models like OPT, LLaMA-2 & 3, Qwen, Mistral across various batch sizes and sequence lengths without compromising accuracy. To our knowledge, this is the first work to demonstrate that contextual sparsity can scale effectively to large batch sizes, delivering substantial inference acceleration with minimal changes, making Polar Sparsity practical for large-scale, high-throughput LLM deployment systems. Our code is available at: https://github.com/susavlsh10/Polar-Sparsity.

## 1 Introduction

Modern LLMs have emerged as powerful tools capable of excelling at diverse tasks, leading to their widespread adoption in modern systems [56, 46, 15, 52]. However, their massive scale, typically involving billions of parameters, makes them computationally expensive and highlights the need for more efficient and cost-effective deployment solutions.

While sparsity and pruning have been extensively studied, their use in production LLMs remains limited due to poor workload generalization and inefficient hardware utilization caused by irregular memory access [26, 61, 55, 59, 37, 58, 35]. A recent line of research has uncovered the phenomenon of contextual activation sparsity, where each input token dynamically activates only a small subset of input-dependent neurons, enabling efficient acceleration without compromising model quality [39]. Activation sparsity emerges as a promising acceleration technique for LLMs with structured sparsity which enables coalesced memory access and measurable wall-clock speedups.

Activation sparsity has been shown to be effective in reducing inference latency, but its application has been largely limited to single-query workloads [39, 54, 4]. Modern inferencing systems depend on batching to maximize hardware efficiency and reduce serving costs [31, 10, 69, 45]. Neural

39th Conference on Neural Information Processing Systems (NeurIPS 2025).

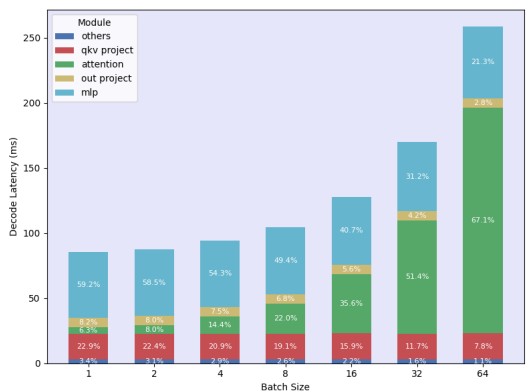
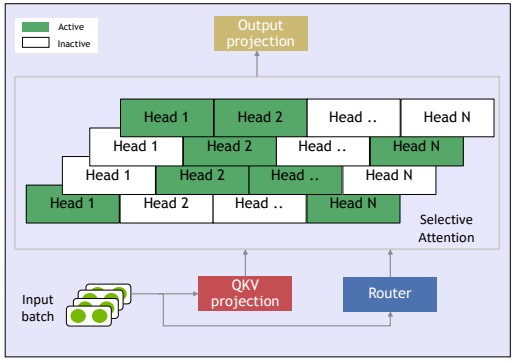

(a) Transformer decode latency breakdown

(b) Selective Head Attention

Figure 1: (a) Decode latency breakdown on A100 GPUs (OPT-66B, sequence length = 1920). As batch size increases, attention layers dominate end-to-end latency. (b) Selective Head Attention only activates the most critical heads for each request and accelerates high throughput inference.

scaling laws have driven the rise of massive models that rely on multi-GPU inference, rendering single-query execution prohibitively expensive for real-world deployment [30, 21, 13]. Recent work on activation sparsity focus on reducing latency by optimizing neuron activations in MLP and linear projection layers as these modules dominate runtime in small batch sizes. However, as shown in Figure 1a, this strategy breaks down under batched workloads. While batching improves the compute efficiency of linear layers, it has the opposite effect on attention, which scales with sequence length and batch size. Furthermore, union neuron sparsity, which varies dynamically with batches and layers, diminishes with batching (Figure 2), limiting the practical gains of existing approaches. This work explores techniques for integrating contextual sparsity into batched inference, enhancing both system throughput and computational efficiency.

Polar Sparsity refers to the shift in sparsity importance from MLP layers to Attention layers as batch size and sequence length increase. Current state-of-the-art sparsity methods primarily focus on model parameter sparsity, where only a subset of model parameters is activated to reduce computation and memory IO. We define head sparsity as the phenomenon where, for a given token, only a subset of attention heads contribute significantly to the output while the remaining heads have negligible effect. In large-batch inference, the cost of accessing model parameters is largely amortized, since the entire batch utilizes the same model weights. In contrast, each batch has a unique key-value (KV) cache, making attention layers memory I/O expensive. While contextual sparsity in model parameters diminishes as batch sizes increase, attention head sparsity remains stable and batch invariant. We introduce Selective Head Attention (Figure 1b), which activates only the most critical heads for each request, preserving overall sparsity and improving compute and memory efficiency.

We leverage these properties to build contextual sparsity-aware Selective GEMM and Selective FlashAttention kernels that reduce memory I/O and compute, enabling scalable and high-throughput inference. To the best of our knowledge, we are the first work to show that contextual sparsity is scalable with batch size, offering even higher gains at larger batch sizes. Our main contributions are as follows:

1. We show that activation sparsity in the MLP layers degrades with batch size due to union activations, while attention head sparsity remains stable and batch-invariant.

2. We design Selective GEMM kernels with a layer-wise top-$k$ optimization strategy for dynamic MLP activations, achieving up to $5.5\times$ speedup.

3. We introduce Selective FlashAttention kernels that support Head/Group sparsity with per-query activation, reducing memory I/O and compute, achieving up to $2.8\times$ speedup.

4. Polar Sparsity delivers up to $2.2\times$ improvement in batched decoding throughput with negligible accuracy loss, and can be seamlessly integrated into a wide range of LLMs, including those without ReLU activations, to unlock substantial inference acceleration.

## 2 Background and Related Works

Contextual sparsity can be leveraged at runtime through sparsity predictors or learned routers that dynamically select the neurons to activate based on the input vectors. Similar to Mixture-of-Experts (MoE) architectures, it activates only a subset of the model conditioned on the input, but differs by operating at a finer granularity, selecting individual neurons rather than routing to large expert blocks.

Early models employing ReLU activations demonstrated high activation sparsity, with over 90% of the feed-forward network outputs being zero-valued, allowing significant computational savings. This indicates that, for a given input, only a small fraction of neurons contribute meaningfully at each decoding step. However, recent LLMs adopt smoother activations like SwiGLU, which yield denser activations and limit the benefits of sparsity. To reintroduce sparsity, several recent studies have proposed returning to ReLU or developing sparse variants [44, 53, 3]. Several subsequent works have been performed to further increase the sparsity by introducing novel sparsity-enhancing techniques and activation functions [33, 37, 68, 50, 66]. Additional discussions can be found in Appendix A.

Several works explore token sparsity in attention, leveraging the fact that many tokens contribute little to model output [14, 47, 22, 1, 62, 40]. This is orthogonal to head sparsity. MoA [65] and MoH [29] introduce MoE-style routing over attention heads, improving accuracy and theoretical efficiency but without wall-clock gains—showing that reduced FLOPs do not always yield faster inference. DejaVu [39] and TEAL [37] also exploit head sparsity but only within the projection layers: they compute QKV vectors for selected heads and copy the corresponding KV cache into smaller buffers, incurring costly memory I/O and limited scalability. Moreover, skipping heads during QKV projection can lead to incomplete context in later decoding. To address these issues, we retain dense QKV projections for KV-cache consistency and apply head sparsity within the attention kernel. We further design an I/O-efficient Selective FlashAttention kernel that scales efficiently to large batches while preserving sparsity benefits. Finally, while prior activation sparsity methods target single-query inference, we extend activation sparsity to the large-batch regime via batch-invariant head sparsity and custom GPU kernels, enabling scalable, high-throughput LLM inference.

## 3 Motivation and Problem Formulation

The inference process in LLMs consists of two stages: prefill and decode. The prefill stage processes the input prompt in a single forward pass to build and cache the KV representations at each transformer layer, and is typically compute-bound. The decode stage then generates tokens autoregressively using the cached KV data; this stage is generally I/O-bound, as each token requires minimal computation but frequent memory access. While prefill has quadratic complexity in sequence length, it is highly parallelizable across tokens and GPUs; by contrast, the decode stage is sequential, I/O-bound, and dominates wall-clock latency in large batch, long-sequence generation. This work focuses on optimizing the autoregressive decode stage under batched inference, aiming to improve throughput and efficiency in real-world serving scenarios.

Figure 1a shows the decode latency breakdown across key modules—MLP, attention, QKV projection, output projection, and others (including communication, layer norm, and embedding projections)—for the OPT-66b model. At small batch sizes, latency is dominated by linear layers, making prior work on activation sparsity highly effective in reducing decoding latency. However, as batch size increases, linear layers benefit from improved compute efficiency, while attention latency grows nearly linearly, quickly becoming the dominant bottleneck. This shift underscores a critical insight: optimizing linear layers alone is insufficient at scale. Our work builds on this observation and focuses on improving decoding efficiency by addressing attention's growing cost in batched autoregressive inference.

### 3.1 Accelerating MLP Layers

In the decode stage, the input to the MLP block is a 3D tensor $x \in \mathbb{R}^{B \times 1 \times d}$, where $B$ is the batch size and $d$ is the model dimension. The two projection matrices in the MLP block are $W_1, W_2 \in \mathbb{R}^{d \times D}$, where $D$ is the hidden dimension of the feed-forward network.

With contextual sparsity, only a subset of neurons in the MLP block are activated for a given input. Let $S \subseteq [D]$ denote the set of active neurons for a single sequence or token in the decode stage. Under batching, we compute the union of active neurons across all sequences in the batch. That is, a

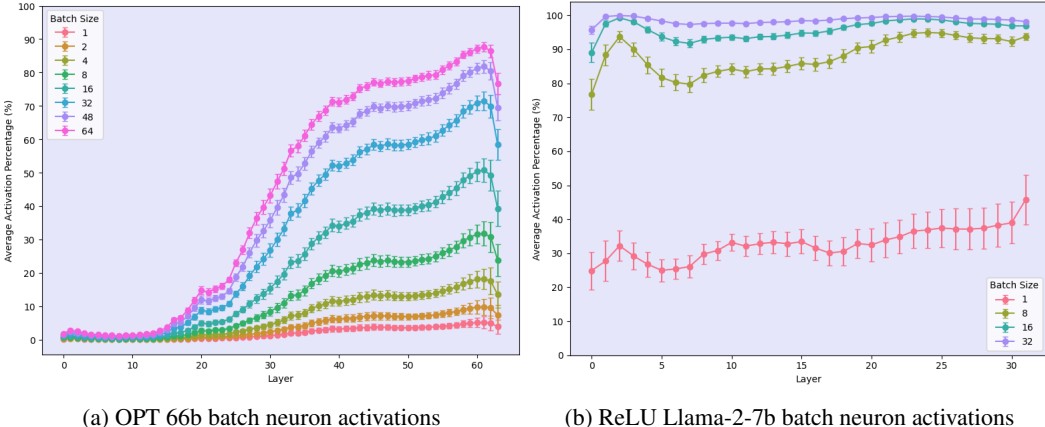

(a) OPT 66b batch neuron activations        (b) ReLU Llama-2-7b batch neuron activations

Figure 2: Batch neuron activations in MLP layers in sparse models. Union neuron sparsity in MLP layers decreases rapidly with batch size, illustrating why MLP sparsity is less effective at scale.

neuron is retained if it is activated for any sequence in the current batch. Let $S_B \subseteq [D]$ denote this union set for batch size $B$. The sparsified MLP computation becomes:

$$\text{MLP}_{S_B}(x) = \sigma\left(xW_{1,S_B}\right)W_{2,S_B}^{\top},$$

where $W_{1,S_B}, W_{2,S_B} \in \mathbb{R}^{d \times |S_B|}$ are the dynamically pruned weight matrices corresponding to the active neurons in the batch, and $\sigma$ is the activation function. As batch size increases, the union $S_B$ typically grows, reducing the overall sparsity and limiting computational savings. Figure 2a shows the average union activation across transformer layers for different batch sizes in the OPT 66b model during the decode stage. We observe a strong layer-wise trend: early layers maintain high sparsity, while deeper layers exhibit progressively higher union activation. Even at large batch sizes, union activation in early layers remains low as the average per-token activation is under 1%. This trend is encouraging, as it reveals an opportunity to accelerate early layers through sparsity, even in large-scale batching. In contrast, sparsity patterns in sparsified LLaMA-2-7B models are more sensitive to batching. Figure 2b shows that replacing SwiGLU with ReLU yields high initial sparsity that quickly diminishes with batching. See Appendix B for additional discussion on MLP sparsity.

## 3.2 Accelerating Attention Layers

In the decode stage, the input to the attention layer is a batch of inputs $x \in \mathbb{R}^{B \times 1 \times d}$, where $B$ is the batch size and $d$ is the model dimension. After applying the QKV projections and reshaping, the query tensor has shape $Q \in \mathbb{R}^{B \times H \times 1 \times d_h}$, while the key and value tensors are updated and retrieved from the KV cache with shape $K, V \in \mathbb{R}^{B \times H_{kv} \times N \times d_h}$, where $H$ is the number of attention heads, $H_{\text{kv}}$ is the number of key-value heads, $d_h = d/H$ is the per-head dimension, and $N$ is the current sequence length of each batch [1]. Each query attends to its own key-value tensor, with self-attention computed independently and in parallel across both heads and batch dimensions.

For each attention head $i \in [H]$ in each batch $b \in [B]$, the attention is computed as:

$$\text{Attention}(Q_{b,i,:,:}, K_{b,i,:,:}, V_{b,i,:,:}) = \text{softmax}\left(\frac{Q_{b,i,:,:}K_{b,i,:,:}^{\top}}{\sqrt{d_h}}\right)V_{b,i,:,:},$$

Since each sequence in the batch attends over its own KV tensors, the attention computation and memory I/O scale linearly with both batch size and sequence length during decode. Importantly, exploiting head-level sparsity can significantly reduce both compute and memory I/O at scale. Unlike MLP sparsity, this form of sparsity is invariant to batching, as each sequence computes attention independently. Thus, attention head sparsity remains a reliable target for acceleration even at large batch sizes.

---

[1]Some implementations represent the tensors as $Q \in \mathbb{R}^{B \times 1 \times H \times d_h}$ and $K, V \in \mathbb{R}^{B \times N \times H_{kv} \times d_h}$

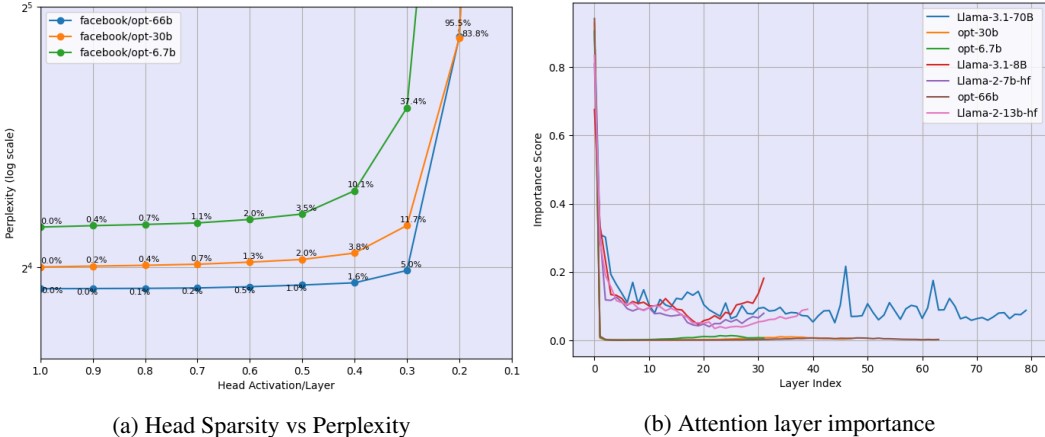

(a) Head Sparsity vs Perplexity        (b) Attention layer importance

Figure 3: (a) Perplexity increases gradually when only activating the most important attention heads/layer, showing that many heads can be skipped with minimal degradation. (b) Layer 0 has high importance score across a range of models, motivating the use of dense attention in early layers.

To study head sparsity, we conduct an experiment where, at each transformer layer, only the top-$k$ attention heads, ranked by output L2 norm, are activated, while the outputs of non-activated heads are masked to zero. We then measure the model's perplexity on a subset of the Wikitext-2 dataset [42]. Figure 3a illustrates the relationship between perplexity and head sparsity, with the relative increase in perplexity annotated at each sparsity level. We observe that by activating the most important heads at each layer, the perplexity does not increase drastically up to 50% head sparsity, across all the tested models. Interestingly, we observe that head sparsity increases with model size. For example, the OPT-66b model shows only a 5% increase in perplexity at 30% head activation, while the smaller OPT-6.7b model experiences a more substantial 37.4% increase. This trend aligns with the expectation that larger models, which possess more attention heads, inherently activate more heads at comparable activation levels. As a result, larger models present a greater opportunity for acceleration through increased sparsity.

Recent work has also highlighted variation in attention importance across layers. We compute per-layer attention importance using the scoring method from [23]. As shown in Figure 3b, layer 0 consistently exhibits the highest importance score across a range of models. Based on this observation, we apply dense attention at layer 0 and enforce uniform head sparsity in all subsequent layers.

## 4 Polar Sparsity

We define **Polar Sparsity** as the emergence of distinct sparsity effectiveness regimes in large-scale LLM inference: MLP sparsity accelerates small-batch, low-latency inference, whereas Attention head sparsity unlocks high-throughput inference at scale. In this section, we present our system for accelerating high-throughput LLM inference via scalable contextual sparsity.

### 4.1 Dynamic Sparsity in MLP Blocks

Our fine-grained MLP router builds on recent advances in sparse inference [39, 54, 4], enabling selective activation at the neuron level. Given a hidden state for a batch, $x \in \mathbb{R}^{B \times 1 \times d}$, the router is implemented as a lightweight two-layer feedforward network with a bottleneck intermediate layer, and we trained it using a binary cross-entropy loss with the AdamW optimizer. Further details are provided in Appendix C.

Prior methods often use static top-$k$ activation thresholds across all layers, which is suboptimal for batched inference due to the dynamic and layer-specific nature of neuron activations that scale with batch size. To address this, we propose a dynamic top-$k$ mechanism that adapts the number of active neurons per layer based on target recall. A simple greedy algorithm (Algorithm 2) selects the minimal top-$k$ neurons per layer to meet a target recall, using the router's output logits and ground-truth activations. We optimize these thresholds offline to maintain high recall (99%) while minimizing

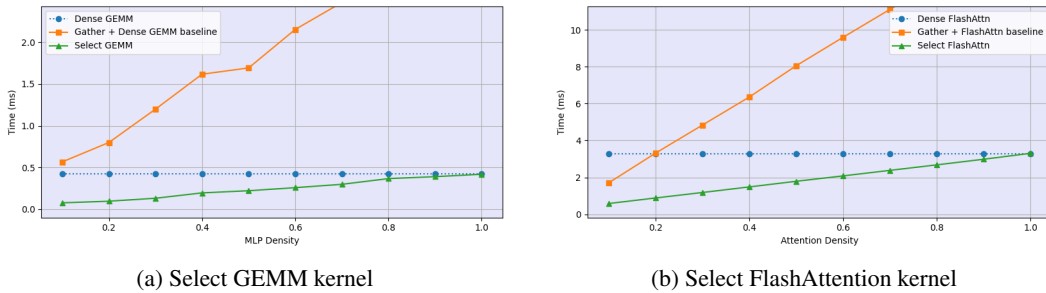

(a) Select GEMM kernel           (b) Select FlashAttention kernel

Figure 4: Polar Sparsity Kernels. Kernels demonstrate near linear speedup with respect to sparsity, A100 GPUs, OPT 66b, batch size 64, seqlen 1920

computation. At inference time, we aggregate the predicted activations across the batch to produce a single *neuron index tensor* that identifies the active neurons for the entire batch.

To efficiently compute sparse MLP activations, we design a custom GPU kernel that fuses indexing and matrix multiplication (Algorithm 3), avoiding the overhead of separate gather-scatter operations. Unlike prior work limited to sparse GEMV, our kernel supports batched GEMM workloads and achieves near-linear speedups with increasing sparsity, with improvements of up to $5.5\times$ over the dense baseline (Figure 4a). Implementation details are provided in Appendix D.

## 4.2 Stable Sparsity with Selective Head Attention

We design our attention routers as single-layer fully connected networks that predict head activations based on attention output norms. The router is trained similarly to the MLP routers: for each input, the top-$k$ attention heads with the highest output norms are considered active and used as supervision targets. The attention router produces scalar logits for each attention head, which are used to select the top-$k$ heads for each batch. Since attention routing is performed independently for each instance, different batches may activate entirely different sets of heads. This results in a *batch head index* tensor that records the active head indices for each batch. Selective Head Attention (SHA) then uses this tensor to perform attention exclusively on the activated heads within a unified kernel.

---

**Algorithm 1** Selective Head FlashAttention (Decode)

---

**Require:** $\mathbf{Q} \in \mathbb{R}^{B \times H \times 1 \times d}$, $\mathbf{K}, \mathbf{V} \in \mathbb{R}^{B \times H \times N_{kv} \times d}$, batch_head_index $\in \mathbb{Z}^{B \times \text{top\_k}}$, $M_{SRAM}, s = 1/\sqrt{d}$
    **Output:** $\mathbf{O} \in \mathbb{R}^{B \times H \times 1 \times d}$ (written to HBM)

1: Determine target batch index $b$ and top-k index $k$ assigned to this unit from the grid dimensions.
2: head_idx $\leftarrow$ batch_head_index$[b, k]$              $\triangleright$ Get the actual head index to compute
3: $B_c = \lfloor M_{SRAM}/(4d) \rfloor$; $(\mathbf{O}_{acc}, l_{acc}, m_{acc}) \leftarrow (\vec{0}, 0, -\infty)$; $T_c = \lceil N_{kv}/B_c \rceil$
4: Load $\mathbf{q} \in \mathbb{R}^{1 \times d}$ from $\mathbf{Q}[b, \text{head\_idx}, 0, :]$       $\triangleright$ Get the activated query vector for the batch
5: **for** $j = 1$ to $T_c$ **do**
6:     $k_{start} = (j-1)B_c$, $k_{end} = \min(jB_c, N_{kv})$; Load $\mathbf{K}_j, \mathbf{V}_j$ from $\mathbf{K}, \mathbf{V}[b, \text{head\_idx}, k_{start} : k_{end}, :]$
7:     $\mathbf{S}_j = s(\mathbf{q}@\mathbf{K}_j^T)$; $\tilde{m}_j = \max(\mathbf{S}_j)$; $\tilde{\mathbf{P}}_j = \exp(\mathbf{S}_j - \tilde{m}_j)$; $\tilde{l}_j = \sum \tilde{\mathbf{P}}_j$
8:     $m_{new} = \max(m_{acc}, \tilde{m}_j)$; $\alpha = e^{m_{acc} - m_{new}}$; $\beta = e^{\tilde{m}_j - m_{new}}$; $l_{new} = \alpha l_{acc} + \beta \tilde{l}_j$
9:     $\mathbf{O}_{acc} \leftarrow (\alpha l_{acc}\mathbf{O}_{acc} + \beta(\tilde{\mathbf{P}}_j@\mathbf{V}_j))/l_{new}$; $l_{acc} \leftarrow l_{new}$; $m_{acc} \leftarrow m_{new}$
10: **end for**
11: Write $\mathbf{O}_{acc}$ to $\mathbf{O}[b, \text{head\_idx}, 0, :]$

---

Given the inputs $\mathbf{Q} \in \mathbb{R}^{B \times H \times 1 \times d_h}$ and $\mathbf{K}, \mathbf{V} \in \mathbb{R}^{B \times H \times N \times d_h}$ in HBM, we aim to compute the attention output $\mathbf{O} \in \mathbb{R}^{B \times H \times 1 \times d}$ for only the activated attention heads for each batch. Our goal is to reduce the amount of memory access and compute by the factor of induced head sparsity. To achieve this, we fuse the head activation logic into a single GPU kernel and perform selective head attention using a sparsity-aware variant of FlashAttention. During batched decoding, each batch and each head is processed in parallel by a different CUDA thread-block or a Triton program. As outlined in Algorithm 1, we modify the FlashAttention algorithm [11] to incorporate head sparsity by indexing into the relevant heads during kernel initialization using a *batch head index* tensor. All memory access logic is updated accordingly to ensure only data from active heads are read from and

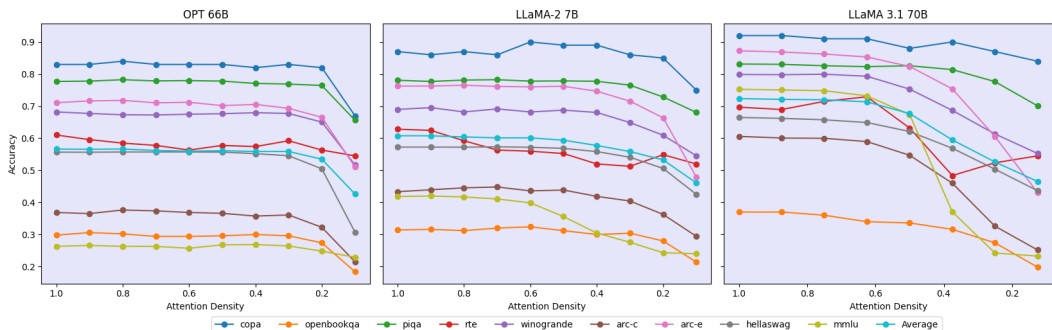

Figure 5: Accuracy vs Attention Density. **Left**: OPT 66b model with dynamic sparse MLP + Select Head Attention. **Middle**: LLaMA 2-7b model with Select Head Attention **Right**: LLaMA 3.1 70b model with Select Group Attention. Dense attention in layer 0 used for all models.

written to, enabling efficient selective head computation within a unified kernel. Figure 4b shows the forward pass runtime of our SHA kernel compared to dense and standard sparse baselines. The SHA kernel exhibits near-linear speedup with respect to head sparsity, achieving $2.8\times$ improvement at 30% sparsity over the dense baseline. For newer models with group query attention (GQA), we embrace group sparsity as query heads within a group share the KV cache [2].

## 5 Evaluation

**Experiment Setting:** We first evaluate *Polar Sparsity* on pretrained models, including OPT [64], LLaMA-2 [57], LLaMA-3 [21], Mistral [28] across multiple model sizes. The evaluation is conducted on nine downstream tasks: COPA [20], OpenBookQA [43], PIQA [7], RTE [19], Winogrande [48], HellaSwag [63], MMLU [24], ARC-easy, ARC-challenging [9]. We utilize the lm-eval-harness framework to measure the accuracy of the models on these benchmarks [17]. We further evaluate *instruction-tuned* models on MMLU-PRO [60] and LongBench [6]. Specifically, we assess LLaMA-3.1-8B-Instruct, Mistral-7B-Instruct, and Qwen-2.5-14B-Instruct [46].

To train the routers, we collected 400,000 token samples from random text sequences extracted from the Wikitext-2 training dataset. All experiments are performed on NVIDIA DGX A100 80GB GPU node servers. Sparsity is applied to both the MLP and attention layers only in OPT models. For all other models, we apply *only attention head sparsity*, since their MLP layers use non-ReLU activations, making MLP sparsification less effective (Appendix B). We scale each model up to batch size of 512 until it reaches the memory limit. We built on top of the FlashAttention triton kernel and utilized CUDA graphs to measure the decoding throughput with the included routers.

### 5.1 Benchmark Evaluation

Figure 5 shows the zero-shot accuracy on downstream tasks as we vary the attention density. At each data point, we activate the attention heads/groups with the highest output logits as predicted by the routers. For OPT models, we also apply dynamic layer-wise sparsity to MLP layers. Across all models, we observe that most tasks can be solved accurately even under high attention head sparsity with minimal degradation up to a critical threshold. The evaluation results show that this critical point varies with the architecture and size of the model. Stable average accuracy is maintained down to 30% attention density for OPT-66b, 50% for smaller OPT-6.7b, LLaMA-2 7b/13b models, and 62.5% for GQA models like LLaMA 3.1 70b.

Table 1 and 2 presents the accuracy of the sparsified models at their respective critical attention density thresholds. Polar Sparsity achieves performance comparable to the dense baseline, with average accuracy differences within 1%. Instruction-tuned models also sustain strong performance on LongBench under Polar Sparsity, underscoring its effectiveness on challenging generative benchmarks.

We observe that some challenging tasks like RTE, MMLU, MMLU PRO are more sensitive to head sparsity and require more active heads. This task-dependent sensitivity is particularly encouraging, as it suggests that most tasks can be served with only the most critical heads, while harder tasks

Table 1: LLM zero-shot evaluation at critical thresholds. Polar Sparsity (PS) is competitive with the dense baseline with average accuracy within 1%.

| Model | COPA | OBQA | PIQA | RTE | WG | HS | MMLU | ARC-E | ARC-C | Average |
|---|---|---|---|---|---|---|---|---|---|---|
| OPT 6.7B | 0.81 | 0.276 | 0.763 | 0.552 | 0.653 | 0.499 | 0.265 | 0.657 | 0.305 | 0.531 |
| OPT 6.7B + PS-0.5 | 0.83 | 0.282 | 0.755 | 0.527 | 0.636 | 0.488 | 0.252 | 0.647 | 0.300 | 0.524 |
| OPT 66B | 0.85 | 0.304 | 0.787 | 0.603 | 0.690 | 0.557 | 0.263 | 0.711 | 0.369 | 0.570 |
| OPT 66B + PS-0.3 | 0.83 | 0.296 | 0.769 | 0.592 | 0.677 | 0.546 | 0.264 | 0.693 | 0.361 | 0.560 |
| LLaMA 2 7B | 0.87 | 0.314 | 0.781 | 0.628 | 0.690 | 0.572 | 0.418 | 0.763 | 0.433 | 0.608 |
| LLaMA 2 7B + PS-0.5 | 0.89 | 0.312 | 0.779 | 0.552 | 0.687 | 0.568 | 0.356 | 0.762 | 0.439 | 0.594 |
| LLaMA 2 13B | 0.91 | 0.350 | 0.791 | 0.653 | 0.722 | 0.600 | 0.521 | 0.794 | 0.485 | 0.647 |
| LLaMA 2 13B + PS-0.5 | 0.92 | 0.352 | 0.790 | 0.578 | 0.728 | 0.600 | 0.473 | 0.783 | 0.473 | 0.633 |
| LLaMA 3.1 70B | 0.92 | 0.370 | 0.831 | 0.697 | 0.799 | 0.665 | 0.753 | 0.872 | 0.606 | 0.724 |
| LLaMA 3.1 70B + PS-0.625 | 0.91 | 0.340 | 0.823 | 0.729 | 0.793 | 0.650 | 0.732 | 0.853 | 0.590 | 0.712 |
| Mistral 7B | 0.92 | 0.332 | 0.803 | 0.686 | 0.738 | 0.609 | 0.591 | 0.796 | 0.489 | 0.663 |
| Mistral 7B + PS-0.5 | 0.92 | 0.340 | 0.801 | 0.671 | 0.736 | 0.608 | 0.562 | 0.793 | 0.483 | 0.657 |

Table 2: Evaluation of instruction-tuned LLMs at critical sparsity thresholds. Polar Sparsity maintains strong performance on generative tasks.

| Model | MMLU PRO | LongBench-e |
|---|---|---|
| Mistral-7B-inst | 0.247 | 0.392 |
| Mistral-7B-inst + PolarSparse-0.5 | 0.244 | 0.388 |
| Llama-3.1-8B-inst | 0.409 | 0.443 |
| Llama-3.1-8B-inst + PolarSparse-0.625 | 0.384 | 0.429 |
| Qwen-2.5-14B-inst | 0.497 | 0.421 |
| Qwen-2.5-14B-inst + PolarSparse-0.625 | 0.457 | 0.414 |

could potentially activate more for higher accuracy within the same batch. Given that head sparsity is batch-invariant, this opens the door to context-aware, per-sequence head activation, paving the way for lossless sparse inference, a direction we discuss in future work.

Although recent activation sparsity literature vary in their model and benchmark selection, we found that several recent methods include LLaMA-2-7b for evaluation. This allows us to perform a fair comparison in Table 3. Polar Sparsity outperforms or is competitive with the state-of-the-art activation sparsity methods across most benchmarks. Importantly, our approach maintains accuracy and scales efficiently with batch size — a key limitation of existing techniques.

Table 3: Benchmark results on LLaMA-2-7b using different sparsity approaches. Zero-shot evaluation unless noted. "–" indicates the metric was not reported in the original paper. Zero-shot OpenBookQA, RTE are unreported in prior work. Bolded numbers indicate the highest among sparse methods. Underlined values denote accuracy within 1% of the dense baseline, competitive performance.

| Method | COPA | PIQA | Winogrande | HellaSwag | MMLU (5-shot) | ARC-e | ARC-c |
|---|---|---|---|---|---|---|---|
| Dense baseline | 0.87 | 0.781 | 0.690 | 0.572 | 0.458 | 0.763 | 0.433 |
| ProbePruning-40% [32] | – | 0.749 | 0.575 | – | – | 0.617 | 0.355 |
| ReLUfication [44] | 0.83 | **0.779** | _0.686_ | 0.548 | 0.386 | 0.738 | 0.396 |
| ProSparse [53] | 0.77 | 0.757 | 0.640 | – | **_0.455_** | – | – |
| CATS-50% [33] | – | 0.769 | 0.675 | **0.571** | 0.421 | 0.744 | 0.412 |
| TEAL-50% [37] | – | _0.778_ | 0.673 | – | 0.405 | – | – |
| GRIFFIN-50% [16] | 0.86 | _0.778_ | – | **0.571** | – | 0.745 | _0.428_ |
| R-Sparse-50% [67] | – | _0.773_ | 0.674 | 0.543 | – | 0.746 | 0.408 |
| PolarSparse-50% | **_0.89_** | **_0.779_** | **_0.687_** | _0.568_ | 0.381 | **_0.762_** | **_0.439_** |

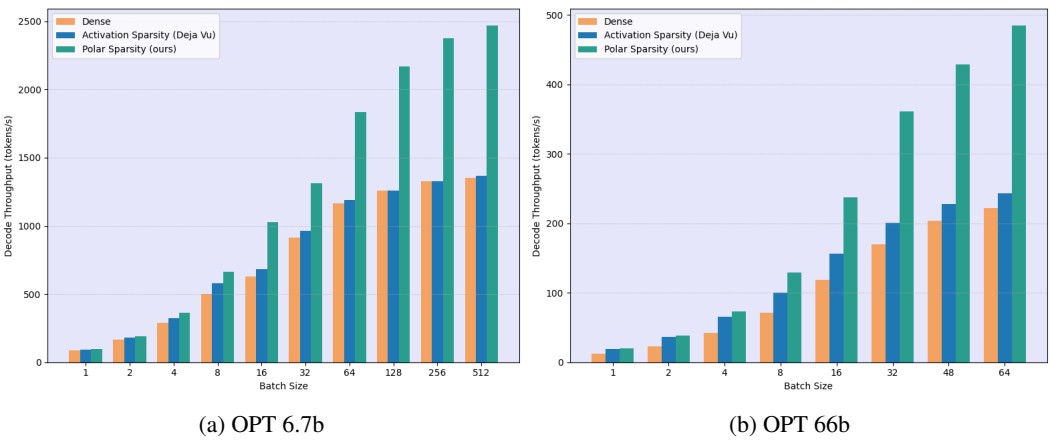

(a) OPT 6.7b                                              (b) OPT 66b

Figure 6: OPT models sparse decoding throughput, seq len 1920 using pipeline parallelism. (a) OPT 6.7b, critical threshold 50%. (b) OPT 66b, critical threshold 30%. Polar Sparsity delivers up to $2.2\times$ higher throughput than dense and up to $2\times$ than standard activation sparsity at scale.

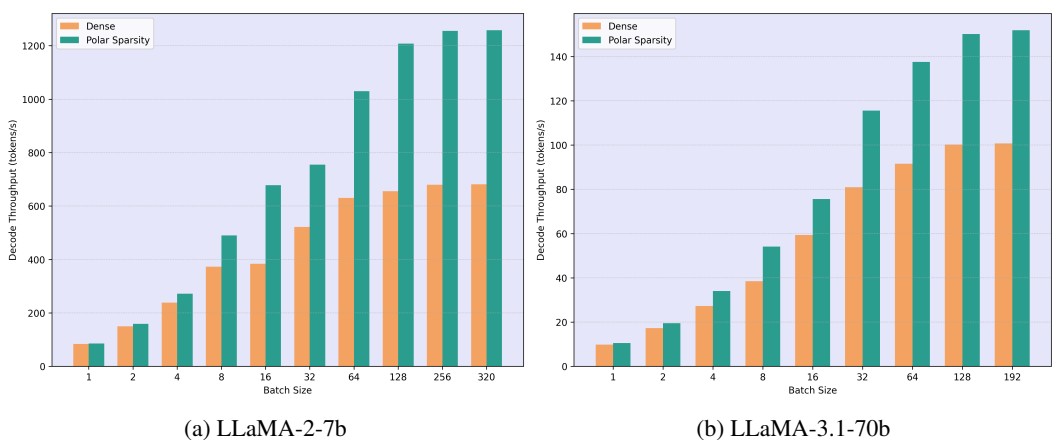

(a) LLaMA-2-7b                                            (b) LLaMA-3.1-70b

Figure 7: LLaMA models sparse decode throughput results using pipeline parallelism. (a) LLaMA-2-7b seq len 3968, critical threshold 50%.(b) LLaMA-3.1-70b seq len 8192, critical threshold 62.5%. Polar Sparsity delivers up to $1.85\times$ higher throughput than dense baseline at scale.

## 5.2 Generation Throughput

In this section, we present the decoding throughput of various models at different batch sizes and sequence lengths at their respective critical threshold. We used sequence lengths of 1920 for OPT, 3968 for LLaMA-2, and 8192 for LLaMA-3 to evaluate sparsity and system performance on increasing workload scales. Additional results for varying sequence lengths are in Appendix E.2.

Figure 6 shows the throughput results of OPT models using Deja Vu-style [39] activation sparsity and Polar Sparsity with the dense baseline at different batch sizes. We substitute DejaVu's sparse GEMV with our Selective GEMM with dynamic top-$k$ to support efficient batched execution. At batch size 1, Polar Sparsity performance is similar to that of conventional activation sparsity. As batch size increases, the effectiveness of conventional activation sparsity diminishes due to the reduced union sparsity across sequences. In contrast, Polar Sparsity leverages the stable, batch-invariant sparsity of attention heads and scales efficiently. Polar Sparsity delivers up to $2.2\times$ higher decoding throughput than dense and $2\times$ more than DejaVu at large batch sizes. In LLaMA-2 models, where sparsity is applied only to attention layers, speedups become apparent at larger batch sizes, reaching up to $1.85\times$. Figure 7b presents throughput results for LLaMA 3.1 70B. Despite higher attention density (62.5%), Polar Sparsity achieves a $1.51\times$ speedup at large batch sizes. While we primarily report

throughput, Polar Sparsity naturally reduces inter-token latency by a similar margin and accelerates per-query generation. Appendix E provides a detailed evaluation across models and workloads.

## 6    Limitations and Future Work

Polar Sparsity is most effective in large-scale, high-throughput settings, particularly during batched decoding. Its benefits diminish for small-batch inference due to limited GPU workload and reduced parallelism. Additionally, we fix the top-k activated heads/groups per layer, whereas a dynamic, input- or layer-adaptive strategy could yield better efficiency and accuracy. Head sparsity could also be combined with token sparsity for potential multiplicative gains. Our experiments use benchmarks with context lengths up to 16k tokens. As newer LLMs reach million-token contexts, extending our analysis to these ultra-long settings is an important direction for future work. We expect our approach to provide efficiency gains in attention variants like Multi-Query [49] and Multi-Latent Attention [12]; however, the critical threshold could be higher similar to GQA. GQA models have naturally less KV diversity in the attention layers and group sparsity is also inherently weaker compared to head sparsity, as it may overlook important heads that reside in inactive groups. This limitation likely contributes to the faster accuracy degradation observed in GQA models. Our current evaluation focuses on greedy decoding. Extending Polar Sparsity to beam search and speculative decoding, where sparsity patterns may differ, offers an exciting direction for future research.

Our approach maintains average accuracy within 1% of the original model at the critical threshold, and this minor drop could potentially be recovered through targeted fine-tuning. Further gains may be possible by training models to induce higher attention head or group sparsity. Since head/group sparsity is batch invariant, there is a promising opportunity to explore task-aware query-sensitive routing, allocating more heads to harder queries and fewer to easier ones within the same batch. Moreover, decoding difficulty can vary across tokens, even within a single sequence. Some tokens may be predicted with fewer active heads, allowing higher sparsity to be applied adaptively at each decoding step. A fine-grained router that dynamically selects heads based on context and difficulty could unlock lossless sparse inference with even greater throughput. We see this as a promising direction toward more adaptive, efficient, and task-aware LLM inference systems.

## 7    Conclusion

Our work highlights the scalability and effectiveness of contextual sparsity for accelerating batched LLM inference. We introduce Polar Sparsity, a key insight showing that as batch size and sequence length grow, the importance of sparsity transitions from MLP layers, where union activation increases, to Attention layers, where head-level sparsity remains stable and batch-invariant.To exploit this property, we develop Selective Head Attention with sparsity-aware GPU kernels that execute computations only for activated heads and neurons. Together, these optimizations deliver consistent speedups across a wide range of models, batch sizes, and sequence lengths with minimal impact on accuracy. Our results are competitive with state-of-the-art approaches and delivers up to $2.2\times$ end-to-end speedups in large-scale settings, affirming the practical viability of Polar Sparsity for efficient and scalable LLM serving. This method is a step towards realizing scalable, high-performance, batched LLM inference that meets the growing demands of modern applications.

## Acknowledgments

This work was performed under the auspices of the U.S. Department of Energy by Lawrence Livermore National Laboratory under Contract DE-AC52-07NA27344 (LLNL-CONF-2005579). This research was also supported in part by the National Science Foundation under Grant No. 2203033. A portion of this work was conducted during an internship at NVIDIA.

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

# A   Extended Related Works

Inference efficiency has been a central focus in machine learning systems, with a diverse set of strategies such as sparsity, pruning, quantization, distillation, speculative decoding, SSD-based offloading, and advanced memory management techniques proposed to reduce computational cost, memory footprint, and latency [8, 38, 27, 34, 18, 25, 51, 31]. These techniques address distinct bottlenecks and are often complementary, enabling combined use for greater gains.

## A.1   Contextual Activation Sparsity

Recent work on activation sparsity in large language models has explored a variety of mechanisms to reduce inference costs without sacrificing accuracy. Early studies revealed that only a small fraction of neurons in transformer MLP layers are active for any given input, establishing the foundation for sparsity-aware inference [36]. Building on this, *ReLUfication* and *ProSparse* showed that replacing GELU or SwiGLU with ReLU or progressively encouraging ReLU-like behavior exposes inherent sparsity and improves inference efficiency, even without retraining [44, 53].

Training-free methods such as *TEAL* and *R-Sparse* apply post-hoc magnitude-based pruning across models like LLaMA-2 and LLaMA-3 and preserve accuracy [37, 67]. *CATS* improves on this by making the threshold contextually adaptive, dynamically pruning low-activation neurons based on input features [33]. *Deja Vu* and *ShadowLLM* train lightweight predictors to anticipate important neurons and attention heads token-by-token, enabling dynamic sparsity at inference time [39, 3]. Our work draws inspiration from previous predictive approaches. Complementary to predictor-based sparsity, *LTE* and *GRIFFIN* optimize structured sparsity patterns during pretraining or in a prompt-aware manner, enabling the model to internalize which parts of the architecture to use for different inputs [68, 16]. Finally, hardware-aware methods like *PowerInfer* leverage the heavy-tailed distribution of neuron activations to map "hot" neurons onto GPU memory while streaming "cold" ones from the CPU, delivering efficient offloading based inference [54].

Existing methods for improving batch efficiency using contextual sparsity are limited in scope. PowerInfer considers batch sizes up to 32 but targets CPU-based offloading systems, where the baseline is already constrained by CPU and PCIe bandwidth [54]. Herd attempts to group sequences with similar sparsity patterns to increase overlap [5], but operates at batch sizes $\leq 4$ and is difficult to apply in dynamic, real-world random queries, where identifying on-the-fly sequences that share similar activation patterns becomes prohibitively challenging. These approaches sparsify only the MLP block and speedup decreases with larger batch sizes. In contrast, our method handles arbitrary sequences and scales effectively to large batch sizes, aligning with realistic deployment scenarios.

## A.2   Attention Head Sparsity and Token Sparsity

Several recent works explore token sparsity to reduce KV cache and attention computation during generative inference. HashAttention introduces semantic sparsity by mapping tokens to a learned Hamming space, enabling efficient selection of pivotal tokens using hash collisions [14]. A2SF proposes accumulative attention scoring with a forgetting factor to fairly rank token importance across time steps in autoregressive decoders [47]. Keyformer and VATP both address token pruning by extending beyond attention scores: Keyformer selects a small set of "key tokens" based on attention concentration [1] , while VATP incorporates the value vector norm into token importance estimation [22]. Native Sparse Attention introduces a hardware-aligned sparse attention mechanism with trainable sparsity patterns suited for long-sequence modeling [62], and MoBA applies mixture-of-expert routing principles to attention tokens by dynamically selecting relevant blocks for each query [40]. These approaches are largely orthogonal to our method and can potentially be combined with Polar Sparsity for multiplicative gains.

Two recent methods, Mixture of Attention Heads (MoA) [65] and MoH: Mixture-of-Head Attention [29], treat attention heads as experts and learn routers to activate a subset per token. MoA requires full model training from scratch, while MoH involves extensive fine-tuning of pre-trained models to learn dynamic head weighting. Although both demonstrate accuracy improvements, they offer limited efficiency gains due to reliance on dense attention kernels. In contrast, our method keeps the model backbone fixed, trains only lightweight routers for head activation, and leverages our custom Select Head Attention kernel to achieve real wall-clock speedups. We believe prior work was limited

by the absence of efficient sparse attention primitives, and with the integration of our kernel, these and future approaches have the potential to achieve both high accuracy and meaningful efficiency gains. We hope this unlocks new opportunities and renews momentum in head sparsity research for accelerating LLM inference.

# B  Extended Study on Batch Activation

## B.1  MLP Activations

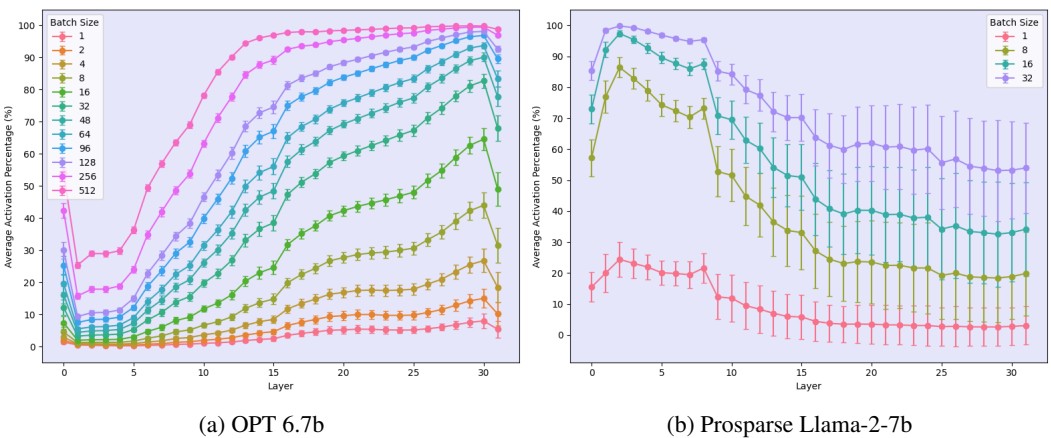

(a) OPT 6.7b

(b) Prosparse Llama-2-7b

Figure 8: OPT and Prosparse Llama Neuron Batch Activations

To better understand the behavior of activation sparsity under batching, we conduct an empirical study across multiple models and sparsification strategies. In this study, a neuron is considered active if its output is greater than zero. We collect true MLP activation statistics by running a forward pass over randomly selected samples from the WikiText-2 dataset. For each model, we group the samples into batches of varying sizes and compute the average union neuron activation and its standard deviation for each layer.

OPT models, which use ReLU activations, exhibit strong inherent sparsity. As shown in Figure 8a, early layers are significantly sparser than deeper ones, a trend consistent across all OPT models. While increasing batch size reduces overall sparsity, the early layers retain enough sparsity to benefit from selective execution. To test the limits of sparsity, we simulated a batch of 10,000 samples and observed that the activations in the initial layers do approach dense and are not, in fact, dead neurons. However, for practical batch sizes, these layers are sufficiently sparse to support accelerated inference. Recent work has shown that sparsity in ReLU-based models tends to increase during pretraining [41]. The OPT models, having undergone extensive large-scale pretraining, likely benefit from this effect, resulting in more structured and persistent activation sparsity.

Figure 8b shows that Prosparse introduces deeper-layer sparsity, but with high variance and similarly degrading trends at larger batch sizes. Unlike OPT, these Sparse-LLaMA variants are only lightly fine-tuned on smaller datasets, which likely limits the emergence of stable sparsity patterns observed in fully pretrained models. Given these observations, we focus our selective execution in LLaMA models on attention head sparsity, where the activation structure remains more consistent across batches.

## B.2  Attention Head Activations

Figure 9 presents heat maps of the head activation counts for the attention heads across all layers in the OPT 6.7B and Llama 2 7B models on 100,000 random token samples from the WikiText-2 dataset. The visualizations reveal that activation patterns vary substantially both from layer to layer and from head to head. In both architectures, some heads are activated significantly more often than others, leading to a skewed distribution of usage. Most layers show a relatively uniform spread of activations across all heads. These findings suggest that future work should focus on designing dynamic head

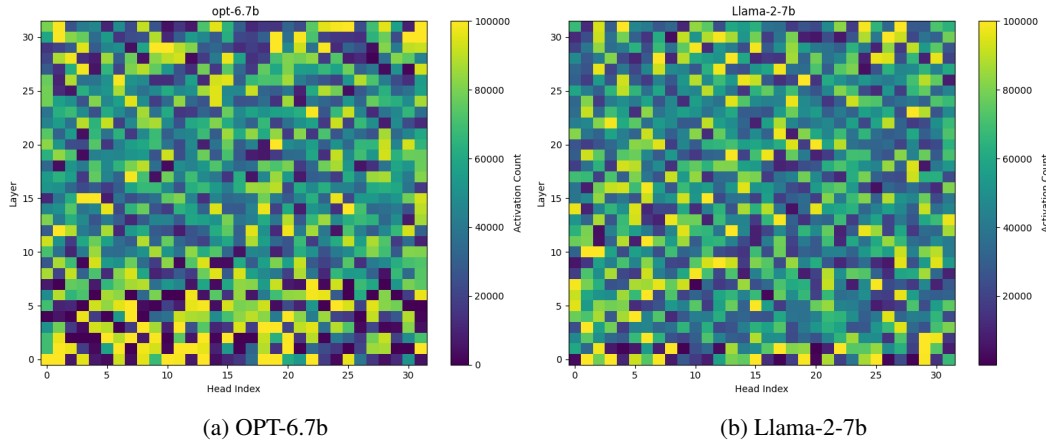

(a) OPT-6.7b         (b) Llama-2-7b

Figure 9: Head activation heat map

allocation strategies that adapt to each layer's unique activation and importance profile, rather than relying on a fixed global threshold.

## C  Sparsity Prediction

Our MLP router follows the design used in prior contextual sparsity work [39, 54]. Each router is a two-layer feedforward network with a hidden dimension of 1024, trained independently for each transformer layer. To partially hide the latency of the MLP router, we overlap its execution with the Attention layer. It is trained using supervised learning with ground-truth neuron activations derived from dense MLP outputs. At inference time, the router produces scalar logits for each neuron, which are used to rank and select the top-$k$ neurons for activation. The attention head routers use a single layer feedforward network as described in Section 4.2. Since the attention routers are much smaller than the MLP routers, we simply run them synchronously before each attention layer. The routers are optimized as binary classifiers using a binary cross-entropy loss, with the AdamW optimizer. We use a batch size of 64, a learning rate of 1e-4, and early stopping over a maximum of 20 epochs. The LLM parameters are frozen during router training. Supervision data is collected from inference runs on the WikiText-2 dataset, as described in Section 5. To determine minimal top-k values for the MLP layers, we apply a simple greedy algorithm (Algorithm 2) that incrementally adjusts the threshold to meet the target recall of 99%. This calibration is performed offline for each OPT model variant.

---

**Algorithm 2** Greedy Top-$k$ Selection to Meet Target Recall

---

**Require:** $R$ (router), $H$ (hidden states), $T$ (true activations), $k_0$ (initial top-$k$), $r_{\text{target}}$ (target recall), $\delta$ (step size)
1: $k \leftarrow k_0$
2: $r \leftarrow 0$
3: **while** $r < r_{\text{target}}$ **do**
4:      $\hat{A} \leftarrow R(H)$                                               ▷ Predict activations
5:      $r \leftarrow \text{COMPUTERECALL}(\hat{A}, T, k)$
6:      $k \leftarrow k + \delta$
7: **end while**
8: **return** $k$

---

### C.1  Router Ablation Study

Figure 10 shows the decode latency of MLP and Attention blocks along with their respective routers, and includes the measured inference latency of sparse kernels across selected sparsity levels. The MLP router introduces approximately four times higher latency compared to the attention router. At higher activation levels, the combined latency of the MLP router and sparse MLP approaches,

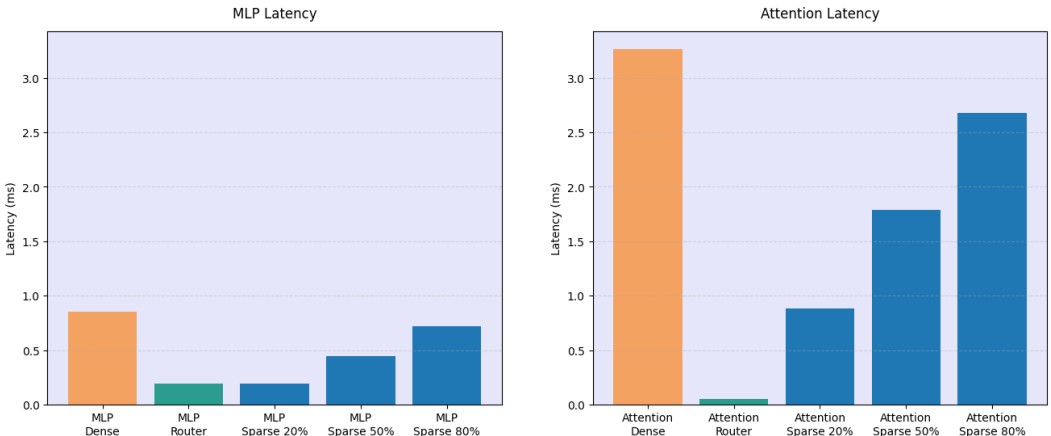

Figure 10: OPT 66b, MLP and Attention Decode Latency with Router at Different Sparsity Levels. Batch size 64, Seq len 1920. MLP router 4 × more expensive than Attention router.

and in some cases exceeds, that of the dense MLP. To address this, we overlap the MLP router execution with the attention block, following a strategy similar to Deja Vu [39], in order to hide its cost. However, under batched inference, it is not always possible to fully mask the MLP router latency due to high GPU utilization. In practice, we observe that overlapping saves approximately 0.1 ms of router latency.

In contrast, the attention router is significantly smaller and more efficient, enabling synchronous execution without incurring high overhead. This results in substantial performance gains: at 50% head activation, the attention block achieves a latency reduction of approximately 1.4 ms, even after including the router cost. Under similar conditions, the MLP router yields only a modest improvement of around 0.21 ms. This further highlights the scalability of head sparsity for large batch workloads.

## D    Sparse Kernels

Standard implementations of selective matrix multiplication typically involve separate indexing of $W_{1,S_B}, W_{2,S_B}$ followed by dense GEMM operations, introducing unnecessary memory overhead. To avoid this, we fuse the indexing and GEMM into a single kernel that dynamically processes only the activated neurons specified by the neuron index tensor as shown in Algorithm 3. To ensure coalesced memory access, we store the weight matrices with the neuron dimension, $d$, as the innermost (contiguous) dimension. Unlike prior work that often targets sparse matrix-vector (GEMV) operations, our kernel is optimized for matrix-matrix (GEMM) operations, enabling efficient execution for arbitrary batch sizes.

---

**Algorithm 3** Sparse Fused GEMM Kernel

---

1: **Input:** $A \in \mathbb{R}^{M \times K}$, $B \in \mathbb{R}^{K \times N}$, index vector $I$
2: **Output:** $C \in \mathbb{R}^{M \times |I|}$
3: **Initialize** thread IDs, memory offsets
4: **Gather** indexed columns or rows of $B$ using $I$
5: **Compute** $C = A \times B_{selected}$ via block-wise multiplication
6: **Apply** activation function (e.g., ReLU)
7: **Store** result back to memory

---

Figure 11 demonstrates that our selective MLP and attention kernels exhibit strong performance and deliver consistent speedups across NVIDIA GPU families (A5000, A100, and H100). As MLP and Attention density increases, latency scales near linearly, indicating that the kernels maintain high efficiency and effectively exploit sparsity without incurring significant overheads.

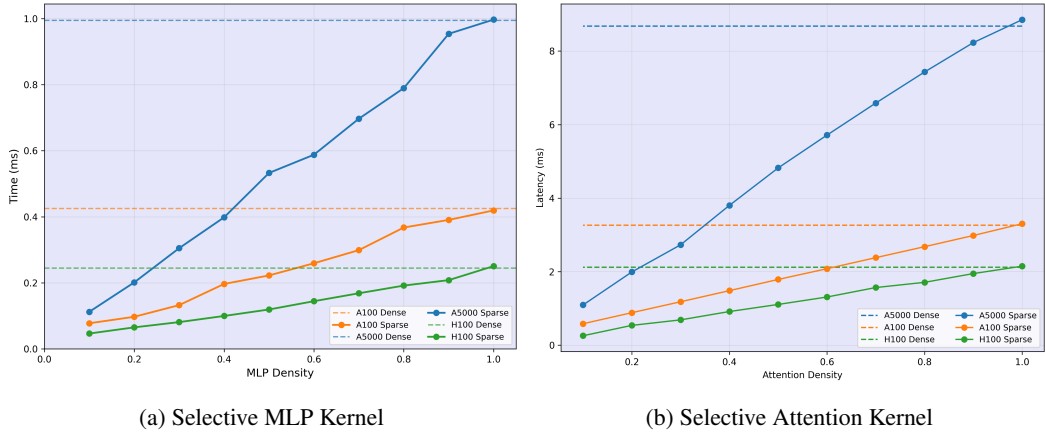

(a) Selective MLP Kernel        (b) Selective Attention Kernel

Figure 11: Selective MLP and Attention Kernel Performance across GPU Families. OPT 66b, batch size 64, sequence length 1920.

# E   Extended Throughput and Latency Evaluation

## E.1   Pipeline Parallel Execution

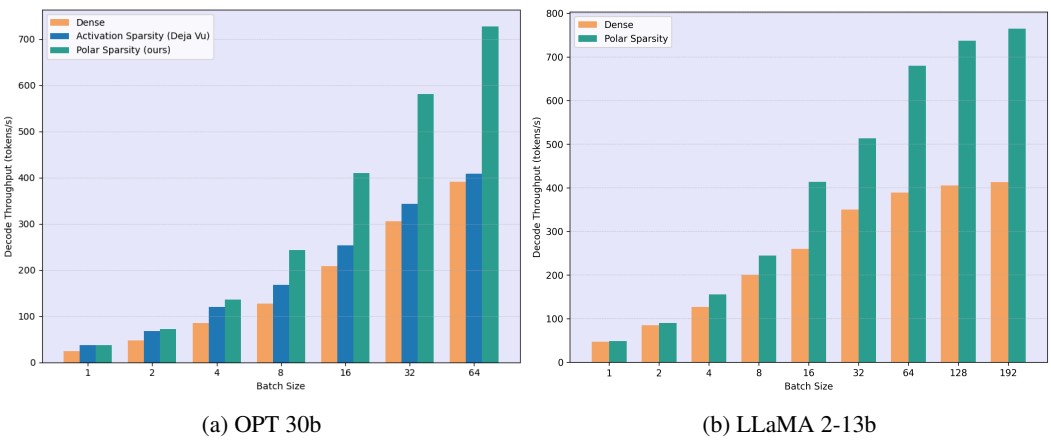

(a) OPT 30b            (b) LLaMA 2-13b

Figure 12: Additional Sparse Decoding Throughput Results. (a) OPT 30b critical attention density 40%, seq len 1920 (b) LLaMA-2-13b attention density 50%, seq len 3968

In this section, we present our results using a pipeline parallel setup without micro-batching. Pipeline parallelism is a technique that divides a large model into sequential stages, distributing the computation across multiple GPUs to improve memory efficiency and throughput, and it is the most commonly used technique for inferencing.

Figure 6, 12a shows the decoding throughput for the OPT model family. For the OPT-6.7B model, despite around 90% sparsity in the MLP layer, the speedup is modest for small batch sizes due to insufficient workload to fully utilize the GPU's parallel capabilities, highlighting the diminishing returns of sparsity in smaller models when the GPU is already underutilized. However, at larger batch sizes, speedups increase to $1.83\times$ as attention computation becomes more memory-intensive. In contrast, the larger OPT-66B model shows more significant improvements, with speedups ranging from $1.66\times$ at batch size 1 to $2.2\times$ at batch size 64, indicating better scaling with workload size.

Figure 7, 12b show the decoding throughput and speedup for the Llama model family. Since we sparsify only the attention layers, speedups are limited at smaller batch sizes due to the overhead of the router and the dominance of MLP layers in computational cost. As batch size increases, the computational burden shifts towards the attention layers, making sparsity more effective. For Llama 2 models, this results in speedups of up to $1.85\times$.

### E.1.1 Tensor Parallel Execution

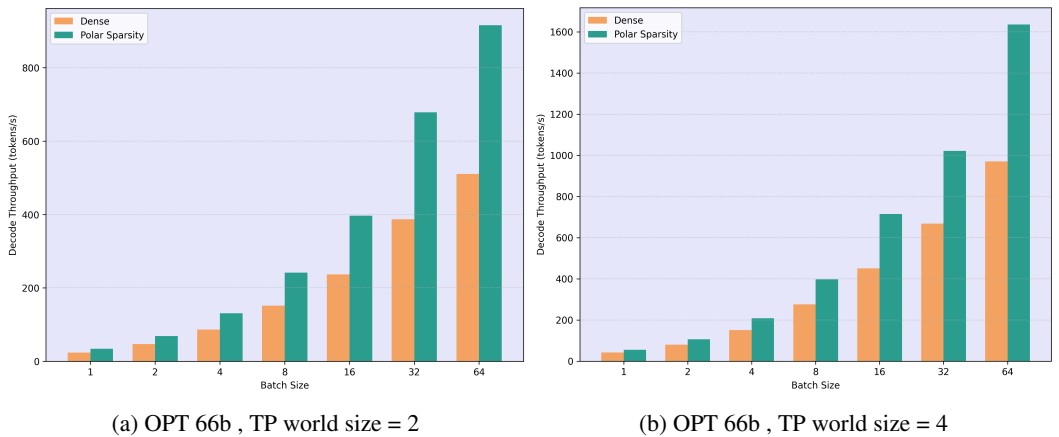

(a) OPT 66b , TP world size = 2          (b) OPT 66b , TP world size = 4

Figure 13: OPT 66b model with Tensor Parallel Sparse Decoding Throughput.

Figure 13 shows the decoding throughput and speedup for the OPT-66B model with tensor parallelism (TP) set to 2 and 4. Tensor parallelism splits individual model layers across multiple devices so each computes a portion of the operation in parallel, reducing memory and computation per device, and is commonly used during training. As the TP degree increases, we observe that the speedup for smaller batch sizes decreases. This is because the workload is divided among multiple GPUs, and applying sparsity further reduces the workload per GPU, leading to diminishing returns when the workload is already small. This is similar to what we observed in the OPT-6.7B model, where sparsity had limited impact at low batch sizes due to under utilization. However, at larger batch sizes, where the computational and memory I/O costs become more significant, Polar Sparsity achieves speedups of up to 1.8x, highlighting its effectiveness in reducing overhead at scale.

## E.2 Decode Latency Analysis

The figure 14 illustrates the decode latency of OPT 6.7b and OPT 66b models using dense, standard activation sparsity (Deja Vu), and our proposed Polar Sparsity methods across varying sequence lengths, with a fixed batch size of 16. Across both model scales, Polar Sparsity consistently achieves lower latency than both Dense and Deja Vu baselines, offering up to $2\times$ speedup over Dense and up to $1.52\times$ over Deja Vu in this workload. The figure 15 show the decode latency of LLaMA-2-7B and LLaMA-3.1-70B models across increasing sequence lengths using dense and Polar Sparsity, with batch size fixed to 16. Polar Sparsity consistently outperforms the dense baseline, achieving up to

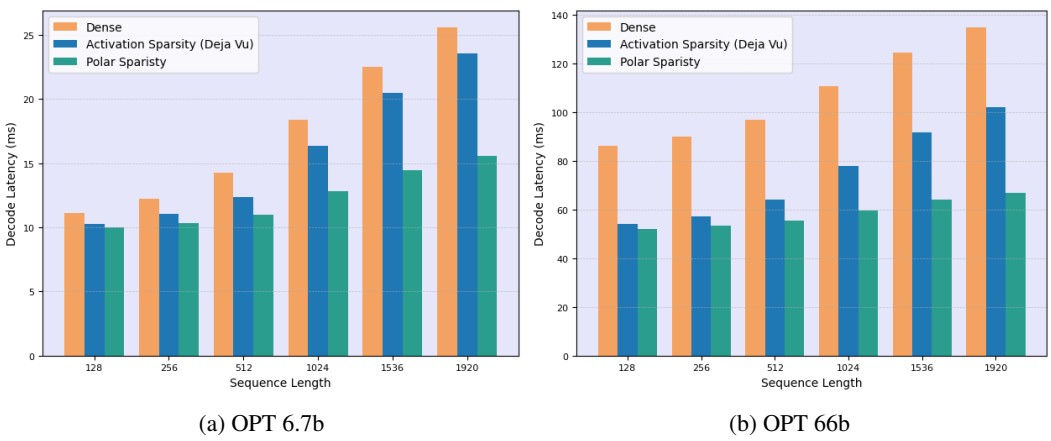

(a) OPT 6.7b                    (b) OPT 66b

Figure 14: OPT models decode, inter-token latency with fixed batch size of 16. Polar Sparsity reduces latency up to $2\times$ compared to dense baseline and up to $1.52\times$ compared to Deja Vu in this workload.

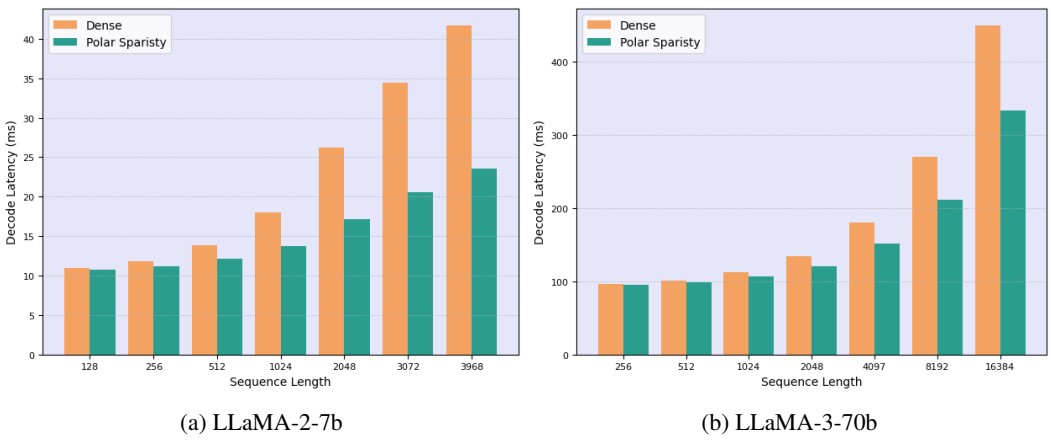

|                    |                    |
| :----------------: | :----------------: |
| (a) LLaMA-2-7b     | (b) LLaMA-3-70b    |

Figure 15: LLaMA models decode, inter-token latency with fixed batch size of 16. Polar Sparsity reduces latency up to $1.77\times$ compared to the dense baseline in this workload.

$1.77\times$ speedup. As with throughput, even greater gains can be expected at larger batch sizes as the attention module dominates latency.

## F  Broader Impacts

Our work improves the efficiency of LLM inference by leveraging contextual activation sparsity, enabling faster and more resource-efficient deployment. This has the potential to make LLMs more accessible in low-resource settings, and support wider adoption in academic and industrial applications. By lowering computational costs, our approach contributes toward more sustainable AI development.

At the same time, accelerating LLM inference can lower the barrier to mass deployment of generative models, which may increase the risk of misuse, such as generating disinformation or harmful content at scale. While our method does not alter model behavior or training data, we recognize that efficiency gains can amplify existing ethical concerns. We encourage responsible deployment practices and safeguards when integrating such techniques into real-world systems.

