# OpenReview forum: "Polar Sparsity: High Throughput Batched LLM Inferencing with Scalable Contextual Sparsity"
_NeurIPS.cc/2025/Conference — NeurIPS 2025 poster_

### Official Review · Reviewer_cRa2 · 2025-06-28

**Clarity:** 3
**Significance:** 2
**Originality:** 2
**Rating:** 5
**Confidence:** 4

**Summary:**

This work proposes an approach for computation reduction at inference time, targeting batch inference via leveraging contextual sparsity in MLP and Attention layers. Given the diminishing gains of sparse MLP execution at large batch sizes, the proposed approach mostly focuses on the attention layers, which are shown to exhibit stable sparsity pattern with the increase of number of samples processed in a single pass. Activated neurons/attention heads are determined via lightweight MLP. And efficient GEMM and Attention kernel is developed and shown to achieve ~2x higher throughput at small or no performance drop. Polar Sparsity is validated on several language models families - OPT, Llama-2 and Llama-3.1.

**Questions:**

- Which framework is adopted for inference? Is the inference benchmark conducted within HuggingFace transformers of some efficient inference engine, such as vLLM or SGLang? If using transformers - which versions of torch/transformers/flash_attention are adopted? The reported speed-ups may vary noticeably depending on a specific version. Are the models compiled prior to execution?
OPT models are very outdated as of 2025. Would be interesting to see, how would the Figure 2 (a) look for newer generations of models, such as Llama-3 or Qwen-2.5/3.
- Wikitext-2 may be not the best source of data for router training. A higher quality data source (such as FineWeb-Edu) may yield better results on benchmarks.
- I think it would be interesting as well to apply Polar Sparsity on a long-context tasks, such as LongBench suite.

[References]
- [1] Penedo, Guilherme, et al. "The fineweb datasets: Decanting the web for the finest text data at scale." Advances in Neural Information Processing Systems 37 (2024): 30811-30849.
- [2] Bai, Yushi, et al. "Longbench: A bilingual, multitask benchmark for long context understanding." arXiv preprint arXiv:2308.14508 (2023).

**Ethical Concerns:**

["NO or VERY MINOR ethics concerns only"]

**Final Justification:**

After reading the rebuttal and the responses addressed to me and the other reviewers, I have decided to raise my score to `Accept`.

The evaluation results demonstrate the robustness of the proposed method across diverse settings. The increase in decoding throughput is quite significant, highlighting the great promise of the method in high‑throughput scenarios.

**Limitations:**

yes

**Paper Formatting Concerns:**

-

**Quality:**

2

**Strengths And Weaknesses:**

**Strengths**
- The training of sparse routers is quite cheap - relatively small amount of data and finetuning budget is sufficient to get good results.
- The provided speed-up is quite decent, given that the accuracy drop is quite small for 50/62.5% sparsity.

**Weaknesses**
* Conceptually, the idea doesn’t appear to be novel. Use of routers for contextual sparsity was previously explored in prior art (referenced DejaVu and Power Infer works). The main contribution of the work is the specific focus on the Attention heads and design of an efficient kernel. I believe that some analysis of the learned attention gates, which heads are activated given some input or some study, that could explain higher drops on MMLU compared to other tasks would strengthen the work.
* The evaluation involves only log-likelihood tasks. The impact of skipping some neurons/heads is likely to be more pronounced in generative tasks. I would propose trying one (or more) of {GSM8K [1], IFEval [2], MMLU-Pro [3]} benchmark. Generative evals are closer to real-world tasks.

[References]
- [1] Cobbe, Karl, et al. "Training verifiers to solve math word problems." arXiv preprint arXiv:2110.14168 (2021).
- [2] Zhou, Jeffrey, et al. "Instruction-following evaluation for large language models." arXiv preprint arXiv:2311.07911 (2023).
- [3] Wang, Yubo, et al. "Mmlu-pro: A more robust and challenging multi-task language understanding benchmark." The Thirty-eight Conference on Neural Information Processing Systems Datasets and Benchmarks Track. 2024.

---

> ### Author Rebuttal · Authors · 2025-07-31
>
> We sincerely thank reviewer for their time and for providing insightful, constructive feedback. We are encouraged that the reviewer recognized the efficiency of our router training and the decent speedups our method provides. We have carefully considered the weaknesses and questions raised and have conducted extensive new experiments that we believe substantially strengthen the paper.
>
> Below, we address each point in detail.
>
> ## Response to Weakness
>
> ### 1. Novelty and Analysis of Attention Gates
> We appreciate the reviewer's perspective on the conceptual novelty. While we build on the established idea of using routers for contextual sparsity (as seen in DejaVu and PowerInfer), we wish to clarify our key, novel contribution:
>
> **Polar Sparsity is the first work to design, build, and demonstrate practical, high-performance GPU kernels (Selective FlashAttention) that convert theoretical attention head sparsity into measured, wall-clock speedups for large-batch, long-sequence inference with up to 2.2x end-to-end throughput improvements.**
>
> Prior work has either focused exclusively on MLP layers, where sparsity benefits diminish with batch size, or has failed to demonstrate tangible end-to-end latency improvements from attention sparsity. Our primary innovation lies in tackling the attention bottleneck directly and providing the efficient implementation necessary to make this approach practical. We have included an extended discussion of related work in Appendix A to further position our contributions.
>
> Importantly, our work provides a foundation for future research to extend head sparsity into a lossless acceleration framework. Our approach has two key properties that enable this: the batch-invariant nature of head sparsity and the use of lightweight routers with a frozen base model. As we propose in our future work section, these properties would allow for a powerful extension where head sparsity is selected dynamically based on query complexity. This would permit different queries within the same batch to activate different numbers of heads, paving the way for a potentially lossless acceleration framework for modern LLMs. Our work provides the foundational kernels that make dynamic, query-aware routing a practical reality.
>
> Regarding the analysis of learned gates, this is an excellent suggestion. We have observed that more challenging reasoning and coding tasks experience a slightly larger accuracy drop, which supports the intuition that difficult queries do require the activation of more attention heads. A deep analysis of which specific heads activate for certain tasks is a fascinating direction for future research. We have provided a preliminary study on head activation in Appendix B.2 and commit to adding further analysis to strengthen the paper.
>
> ### 2. Evaluation on Generative and Long-Context Tasks
>
> This is a critical point, and we agree that evaluating on generative tasks is essential. In response to this and similar feedback from other reviewers, we have conducted extensive new evaluations on several challenging generative and long-context benchmarks on new instruction tuned models.
>
> **New Results on MMLU-PRO, GSM8K, and LongBench**
>
> We evaluated our sparse models on MMLU-PRO and GSM8K. The results show that our method maintains strong performance with only a minor, graceful degradation in accuracy.
>
> | Subject                    | Llama-3.1-8B-Inst_dense | Llama-3.1-8B-Inst_sparse_0.625 | Mistral-7B-Inst_dense | Mistral-7B-Inst_sparse_0.5 |
> |---|---|---|---|---|
> | MMLU_PRO_Biology           | 0.62 | 0.60 | 0.47 | 0.50 |
> | MMLU_PRO_Business          | 0.44 | 0.43 | 0.21 | 0.20 |
> | MMLU_PRO_Chemistry         | 0.26 | 0.24 | 0.12 | 0.11 |
> | MMLU_PRO_Computer Science  | 0.42 | 0.41 | 0.28 | 0.29 |
> | MMLU_PRO_Economics         | 0.53 | 0.51 | 0.34 | 0.35 |
> | MMLU_PRO_Engineering       | 0.24 | 0.27 | 0.16 | 0.15 |
> | MMLU_PRO_Health            | 0.51 | 0.49 | 0.31 | 0.30 |
> | MMLU_PRO_History           | 0.41 | 0.37 | 0.25 | 0.29 |
> | MMLU_PRO_Law               | 0.28 | 0.28 | 0.17 | 0.17 |
> | MMLU_PRO_Math              | 0.39 | 0.35 | 0.17 | 0.17 |
> | MMLU_PRO_Other             | 0.46 | 0.44 | 0.32 | 0.29 |
> | MMLU_PRO_Philosophy        | 0.43 | 0.42 | 0.26 | 0.24 |
> | MMLU_PRO_Physics           | 0.34 | 0.34 | 0.19 | 0.18 |
> | MMLU_PRO_Psychology        | 0.59 | 0.59 | 0.44 | 0.43 |
> | MMLU_PRO Overall Accuracy  | 0.41 | 0.40 | 0.25 | 0.24 |
> | GSM8K                      | 0.75 | 0.71 | 0.34 | 0.32 |
>
> We also tested our models on the LongBench-e benchmark, which specifically evaluates long-form generation. The results demonstrate the stability and robustness of our approach, with no catastrophic failures observed.
>
> | Metric                | Llama-3.1-8B-Inst_dense | Llama-3.1-8B-Inst_sparse_0.5 | Llama-3.1-8B-Inst_sparse_0.625 | Mistral-7B-Inst_dense | Mistral-7B-Inst_sparse_0.5 | Qwen2.5-14B-Inst_dense | Qwen2.5-14B-Inst_sparse_0.5 |
> |---|---|---|---|---|---|---|---|
> | 2wikimqa              | 15.28 | 14.28 | 14.50 | 32.30 | 31.99 | 13.76 | 12.52 |
> | gov_report            | 34.66 | 31.79 | 33.56 | 28.64 | 28.60 | 30.99 | 27.34 |
> | hotpotqa              | 17.50 | 15.17 | 15.46 | 38.71 | 37.11 | 14.08 | 14.35 |
> | lcc                   | 69.56 | 60.72 | 63.70 | 56.78 | 56.11 | 67.90 | 60.41 |
> | multi_news            | 25.45 | 25.08 | 25.56 | 22.60 | 22.18 | 21.53 | 21.70 |
> | multifieldqa_en       | 27.79 | 27.92 | 26.61 | 32.91 | 32.09 | 35.18 | 32.12 |
> | passage_count         | 15.75 | 18.22 | 18.68 | 4.04 | 4.84| 7.64 | 11.05 |
> | passage_retrieval_en  | 97.12 | 94.24 | 95.46 | 31.67 | 37.33 | 89.64 | 94.61 |
> | qasper                | 11.83 | 11.85 | 12.49 | 30.04 | 28.34 | 10.67 | 10.94 |
> | repobench-p           | 55.31 | 45.19 | 48.76 | 47.14 | 41.77 | 47.62 | 40.15 |
> | samsum                | 42.73 | 38.39 | 40.79 | 40.84 | 39.74 | 44.69 | 41.01 |
> | trec                  | 71.00 | 71.67 | 71.67 | 61.33 | 61.33 | 75.00 | 75.00 |
> | triviaqa              | 92.81 | 89.72 | 90.72 | 83.11 | 82.68 | 88.43 | 86.61 |
> | LongBench-e average   | 44.37 | 41.86 | 42.92 | 39.24 | 38.78 | 42.09 | 40.60 |
>
> These new results confirm that Polar Sparsity preserves model capabilities on the very tasks where high-throughput inference is most critical, adding significant weight to our evaluation. We will integrate these findings into the revised manuscript.
>
> ## Response to Questions
>
> ### 1. Inference Framework, Installed version of libraries
>
> We evaluate our kernels and end-to-end generation performance on top of the FlashAttention library (FlashAttention library includes inference and training scripts) with CUDA graphs enabled. We selected FlashAttention library to test our inference system for its ease of integration with the new Selective FlashAttention kernels and its close performance with vLLM. We are actively working on integrating our kernels with future versions of vLLM.
> We have also included the entire code base in the supplemental materials which includes our triton kernels and inference scripts.
> We used fused kernels for efficient layernorms, MLPs but did not use torch.compile in our latency, throughput measurements. This would be a valuable addition which can further reduce the latency and we are happy to update our graphs with the compilation enabled.
>
> PyTorch version: 2.6.0+cu124
> FlashAttention version: 2.6.3
> Transformers version: 4.50.3
>
> ### 2. Figure 2 (a) with newer models
>
> This is a great point. While Figure 2(a) focused on the OPT family, the majority of our evaluations (throughput in Fig 5, 6 and new accuracy benchmarks) feature modern models like Llama2, Llama-3.1, Mistral-7B, and Qwen2.5-14B. These newer models exhibit similar or even more pronounced attention dominance as these models support larger context lengths. We will gladly add a new version of Figure 2(a) with these modern architectures in the revised version of the paper.
>
> ### 3. Wikitext-2 for router training.
> Thank you for this excellent suggestion. We agree that using higher-quality data like FineWeb-Edu could potentially improve router performance and downstream task accuracy. We used Wikitext-2 as a standard, well-established dataset to ensure a reproducible and comparable baseline. Exploring the impact of different training data sources is a valuable direction for future work.
>
> ### 4. Polar Sparsity on a long-context task.
> We completely agree. As shown in our response to your second weakness, we have now benchmarked our method on the LongBench suite. The results demonstrate that our approach is highly effective for long-context scenarios, maintaining robust accuracy while providing significant speedups.
>
> ## Conclusion
>
> We are grateful for the reviewer's sharp questions and valuable suggestions, which have prompted us to significantly strengthen our paper's evaluation. We have provided extensive new benchmarks on challenging generative and long-context tasks, clarified the novelty of our high-performance kernels, and detailed our experimental setup. We believe these additions directly address the concerns raised and more clearly demonstrate the practical impact and significance of Polar Sparsity. We sincerely hope the reviewer will consider these updates favorably and reconsider their rating.

---

> > ### Comment · Reviewer_cRa2 · 2025-08-02
> >
> > Thank you for your response. My concerns were partially addressed. While similar solutions offering some speed-up do exist in prior work (Deja Vu, Power, FlexGen), the framework in its proposed form does possess some novelty.
> >
> > The provided new benchmark results show only a gentle performance drop, and for the reported speed-ups, the method seems to offer a good trade-off between performance and accuracy.
> >
> > Therefore, I have decided to raise my score.

---

> > > ### Author Response · Authors · 2025-08-05
> > >
> > > Thank you for taking the time to re-evaluate our work and for raising your score. We’re glad that the updated benchmarks and clarification of our contributions helped address your concerns. We remain committed to improving this work further and would love to address any remaining questions or concerns to earn your full confidence in the paper. We sincerely appreciate your constructive suggestions and hope the revised version reflects both the novelty and practical impact of our approach.

---

### Official Review · Reviewer_bAJa · 2025-06-30

**Clarity:** 4
**Significance:** 3
**Originality:** 2
**Rating:** 4
**Confidence:** 4

**Summary:**

This paper proposes Polar Sparsity, which sparsifies the MLP and attention mechanism during the LLM decoding stage.

**Questions:**

- In Figure 1, I think the x label is wrong. Isn't it "sequence length" rather than "batch size"? The latency of attention is the only variable on that figure, so I suspect that the label is wrong.
- Can you point out how S_B is constructed? I cannot well understand how an activated neuron is selected for each batched sequence. Is it just a simple top-k of input activation? If so, what does the input statistics look like? Can you make a heatmap of those tensors? And you mentioned layer coherences, but the visualization of that fact is insufficient, as I think. Can you add more statistics or visualization, such as a heatmap or histogram, to them? I feel like this paper is well written about efficiency and implementation, and not very friendly on methodological explanation and statistical visualizations. Just adding two or three figures about statistics that are needed for intuitions might improve the readability of this paper significantly.
- I think the input sequence should not be random for neural activation statistics sampling (Figure 1b). Can you sample it from real data, such as PG19?

**Ethical Concerns:**

["NO or VERY MINOR ethics concerns only"]

**Final Justification:**

I still have a concern about long context Evaluations, because the authors only did the subset of InfBench, and the results are almost same as random performance.

However, it is clear that this method is effective in certain condition, so I will increase my voting to borderline accept.

**Limitations:**

Please refer to the weakness section for more information.

Despite their mention of MQA and MLA as their first limitation, I do not agree that this limitation significantly draws down their effectiveness. I suggest that authors think about another limitation of this work; therefore, it further helps future researchers.

**Paper Formatting Concerns:**

The paper is well formatted.

The supplementary file is extremely well formatted and helpful. I love to see such high-quality implemenation codes, and I strongly encourage authors to submit such kind of codes in future their careers too.

**Quality:**

4

**Strengths And Weaknesses:**

# Strength
1. Very advanced and well-constructed implementation based on OpenAI Triton.
1. Well scales up with larger batch size, thanks to well-constructed block-wise sparsification rather than scalar or fine-grained level sparsification methodologies.

# Weakness
1. Limited evaluation of context length. I think this paper should evaluate their model on a longer context, because serving a longer context is much more challenging and a more effective scenario when we optimize the Transformers (including the attention mechanism). I think this would improve the effectiveness of this method in longer sequence inference, such as RAG and Agentic AI. Moreover, the main assumption of activation sparisity is not also well studied with long-context.
1. Limited evaluation of sparsity. In Figure 4, the plot only deals with attention sparsity, while the methodology can also dynamically sparsify the MLP. Furthermore, I do not agree that "density" is an "efficiency metric", because mostly FLOPs or mathematical asymptotic computational cost does not well match with real-world GPU kernel latency. But I believe authors are able to easily change this to actual latency on various conditions of GPUs (consumer-grade, server-grade, multi-node), while considering their excellent implementation.
1. More performance drawdown happens on a better model. This might be trivial or not, but I think the ideal trend of this should be better model --> better efficiency. However, if we see the MMLU scores in Figure 4, the Llama 3.1 70B shows a more significant drawdown of the MMLU score compared to OPT. OPT maintains its original performance up to 20%, while Llama 3.1 starts to fall from 60% density. I am disappointed about this result, so I hope authors can test on other modes such as Qwen3 or Mistral so we can show that this is only a special phenomenon on Llama 3.1.
1. Only applied to the decoding part. I think we can apply the sparsification method in the prefill stage also, so we can accelerate the speed of long-sequence prefill, which is usually bottlenecked on long-code prompting.

---

> ### Author Rebuttal · Authors · 2025-07-31
>
> We thank the reviewer for the thoughtful and constructive feedback. We are encouraged by the recognition of the implementation quality and scalability of our approach. Below, we provide detailed responses and updates:
>
> ## Response to Weakness
>
> ### 1. Context length evaluation
>
> Thank you for this excellent suggestion. We agree that evaluating on long-context benchmarks is crucial for demonstrating the real-world utility of our method. To address this, we have conducted new experiments on the LongBench-e dataset with several state-of-the-art instruction-tuned models. The full set of results can be viewed in the response to reviewer sVp2.
>
> | Model  / LongBench-e accuracy by context length | 0-4k | 4-8k | 8k+ |
> |---|---|---|---|
> | Llama-3.1-8B-Instruct_dense              | 44.01 | 44.30 | 44.80 |
> | Llama-3.1-8B-Instruct_sparse-0.5         | 42.45 | 42.13 | 41.02 |
> | Llama-3.1-8B-Instruct_sparse-0.625       | 43.35 | 43.00 | 42.41 |
> | Mistral-7B-Instruct-v0.1_dense           | 44.26 | 39.09 | 34.37 |
> | Mistral-7B-Instruct-v0.1_sparse_attn_0.5 | 44.39 | 38.34 | 33.60 |
> | Qwen2.5-14B-Instruct_dense               | 42.45 | 41.51 | 42.31 |
> | Qwen2.5-14B-Instruct_sparse              | 40.46 | 40.47 | 40.87 |
>
> On long-context benchmarks, model quality remains high. At 50% attention sparsity, the average performance drop is minimal across models (e.g., just -0.46 points for Mistral-7B and -1.49 for Qwen2.5-14B). While challenging coding tasks are more sensitive, many other tasks show negligible change. This modest accuracy trade-off is paired with a significant 1.77x speedup in long-context decoding (Appendix E.2). We will also include latency and throughput evaluations for the newly tested models in the revised manuscript.
>
> ### 2. Evaluation of sparsity
>
> We agree that density, theoretical FLOPs are insufficient and that real-world latency is the most meaningful metric. We have used "density" in Figure 4 primarily to illustrate the trade-off between model accuracy and the degree of attention head activation. The actual wall-clock speedups are presented in Figure 3, 5, 6, Appendix E. Using our optimized kernels, we can improve batch decoding up to 2.2x as reported in the original submission.
>
> Regarding MLP sparsity, you are correct that Figure 4 focuses on attention. In those experiments, MLP sparsity was determined dynamically by our algorithm as described in Section 4.1.
>
> To directly address your concern about hardware-specific performance, we have benchmarked our optimized Triton kernels on both consumer (A5000) and server-grade (A100, H100) GPUs.
>
> Attention Kernel Latency (ms) and Speedup (Batch=64, SeqLen=2k, Num Heads = 72, MHA)
> | Attention Density | A5000 Dense | A5000 Sparse | A5000 Speedup | A100 Dense | A100 Sparse | A100 Speedup | H100 Dense | H100 Sparse | H100 Speedup |
> |---|---|---|---|---|---|---|---|---|---|
> | 1.0 | 8.67 | 8.85 | 0.98x | 3.26 | 3.30 | 0.98x | 2.12 | 2.14 | 0.99x |
> | 0.8 | 8.67 | 7.43 | 1.16x | 3.26 | 2.67 | 1.21x | 2.12 | 1.71 | 1.24x |
> | 0.6 | 8.67 | 5.71 | 1.52x | 3.26 | 2.08 | 1.56x | 2.12 | 1.31 | 1.62x |
> | 0.5 | 8.67 | 4.82 | 1.80x | 3.26 | 1.78 | 1.82x | 2.12 | 1.11 | 1.91x |
> | 0.3 | 8.67 | 2.73 | 3.18x | 3.26 | 1.18 | 2.76x | 2.12 | 0.69 | 3.07x |
> | 0.1 | 8.67 | 1.09 | 7.92x | 3.26 | 0.58 | 5.58x | 2.12 | 0.25 | 8.22x |
>
> The results show consistent and significant speedups that scale linearly with sparsity across different GPU architectures. We achieve ~1.8-1.9x speedup at 50% sparsity.
>
> ### 3. Performance drawdown
>
> This is a very insightful point. We agree the ideal trend is for better models to show more resilience to sparsity. As shown in our original Figure 2, this trend holds within a model family (e.g., OPT-66B is more robust than OPT-6.7B). The variance between families (OPT vs. Llama 3.1) likely stems from different architectures and training recipes.
>
> As requested, we evaluated Mistral-7B to see if the Llama 3.1 result was an outlier.
>
> | attn_density | arc_challenge | arc_easy | copa | hellaswag | mmlu | openbookqa | piqa | rte | winogrande | average |
> |---|---|---|---|---|---|---|---|---|---|---|
> | 1 | 0.489 | 0.796 | 0.920 | 0.609 | 0.591 | 0.332 | 0.803 | 0.686 | 0.738 | 0.663 |
> | 0.8 | 0.491 | 0.796 | 0.920 | 0.609 | 0.591 | 0.340 | 0.802 | 0.682 | 0.736 | 0.663 |
> | 0.6 | 0.488 | 0.792 | 0.920 | 0.607 | 0.577 | 0.342 | 0.804 | 0.668 | 0.734 | 0.659 |
> | 0.5 | 0.483 | 0.793 | 0.920 | 0.608 | 0.562 | 0.340 | 0.801 | 0.671 | 0.736 | 0.657 |
> | 0.4 | 0.474 | 0.789 | 0.920 | 0.606 | 0.535 | 0.330 | 0.801 | 0.646 | 0.733 | 0.648 |
>
> Mistral-7B also maintains its performance robustly down to 50% density. The greater drop on harder tasks like MMLU suggests that while most queries benefit from sparsity, the most challenging ones may require higher head activations. This observation reinforces the value of our dynamic, batch-invariant design, which allows future work to build query-sensitive routers that activate more heads only when needed, making our efficient kernels a critical enabler for such techniques. We evaluated additional models on harder tasks like MMLU_PRO, GSM8K and observed similar trends (see response to reviewer nrZF). This does suggest that Llama-3.1 family could be a possible outlier in this data.
>
> We want to emphasize that we only train lightweight routers leaving the base model frozen. Crucially, while any approximation technique can be lossy, our method's configurable and batch-invariant nature is a key strength. It allows different queries in a batch to activate different heads, paving the way for future work on query-sensitive routing to achieve nearly lossless performance. Our work provides the first efficient kernel implementation to make this direction practical. A recent line of work proposed to fine-tune the base model with head sparsity and has been shown to be able to completely recover accuracy [1]. While this work did not manage to show efficiency gains, we strongly believe that future research expanding this line of work will benefit from our optimized kernels and be able to extract wall-clock speedups with minimal to no accuracy loss. We include further discussions in Appendix A.
>
> [1]. MoH: Multi-Head Attention as Mixture-of-Head Attention.
>
> ### 4. Applied only to decoding
>
> We thank the reviewer for this insightful suggestion. The reviewer is correct that applying our sparsification method to the prefill stage is a valuable potential extension of our work.
>
> In this paper, we chose to focus on the decoding stage as it presents a distinct, memory-bandwidth-bound bottleneck, especially during long-sequence generation, which contrasts with the compute-bound nature of the prefill stage. While sparsification could apply here, the performance trade-offs are different, and it may require a fundamentally different approach, kernels to be effective.
>
> We agree that accelerating prefill is an important problem. We will add this to the future work section of our paper and clarify the scope of our current method in the introduction. We appreciate the reviewer for pointing out this promising research direction.
>
> ## Response to Questions
>
> ### 1. Figure 1 Label
>
> The x-axis in Figure 1 is correctly labeled 'batch size'. The figure illustrates a key motivation for our work: during autoregressive decoding, attention latency scales linearly with batch size and context length, while the latency of other components like MLPs does not. The latency of other modules increases, but at a much slower rate.
> This disproportionate scaling makes attention the primary bottleneck at larger batch sizes, which our method directly targets.
>
> ### 2. How is S_B constructed?
>
> S_B is the set of unique active neurons for an entire batch, constructed as follows:
>
> + A small router network predicts an importance score for each neuron for every sequence in the batch.
>
> + For each sequence, we select the top-k neurons based on these scores.
>
> + S_B is then the union of these selected neurons across all sequences in the batch. Therefore, a neuron is used in the computation if it is selected for at least one sequence.
>
> We appreciate the feedback on improving the methodological intuition. We have included some activation statistics in Appendix B which shows the average activation per layer with standard deviation across a range of batch sizes. We will add the requested heatmaps and histograms of neuron activation to the revised manuscript to better illustrate these dynamics. Additionally, we already have included heat maps for head activations in Appendix B.2.
>
> ### 3. Input sequences in Figure 1b
>
> The term "random sequences" was meant to convey that the input queries were sampled from a diverse dataset without regard to semantic similarity, ensuring a varied activation pattern. We will clarify this in the manuscript and use samples from a standard dataset like PG19 for our new visualizations to improve realism and reproducibility.
>
> ## Response to limitations
>
> We use fixed attention head activation/layer to enable clean benchmarking of our architecture. However, results on challenging benchmarks like MMLU suggest that uniform activation is suboptimal. This motivates a dynamic attention head router, analogous to our MLP router, that can allocate more heads to complex queries. Our high-performance kernels make this direction feasible, and we will include this discussion in the revised manuscript.
>
> ## Conclusion
>
> We thank you once again for your time and insightful feedback. We have incorporated extensive new experiments and analyses directly addressing your concerns about evaluation on long contexts, different hardware, and newer models. We are confident that the manuscript has been substantially strengthened by these additions and hope you will view the revised submission favorably.

---

> ### Comment · Reviewer_bAJa · 2025-08-04
> **Response to Author Rebuttal**
>
> Thank you for your detailed response. I was mostly satisfied with the authors' rebuttal. However, I believe the author's response did not fully resolve my concern due to the limited evaluation. Here are my suggestions.
>
> -----
>
> 1. Context length evaluation
>
> I think LongBench-e is not suitable in this case. Both Mistral 0.1 and Llama 3.1 support up to 128K window, but LongBench-e is designed for models that have shorter context windows, such as 8K tokens. I believe OpenBMB/InfiniteBench (up to 384K), Microsoft/RULER (up to 1M), Google-DeepMind/LOFT (up to 1M) can help this evaluation. I believe long context performance evaluation is especially important when we are dealing with SoTA long context LLMs, because usually they are very sensitive to the attention mechanism.
>
> 3. Performance drawdown
>
> I think authors should test a stronger model, such as Qwen3 235B, or GLM4.5, to resolve my concern. My original suggestion was the accuracy on Qwen3 and Mistral (I apologize that I did not specify the model scale). However, I'm glad to see that the trend in the Llama 3.1 has not happened in Mistral v0.1 (which is still a pretty weak model).
>
> -----
>
> I think my concerns are mostly resolved, except for the long-context evaluation and large-scale model evaluation.

---

> > ### Author Response · Authors · 2025-08-05
> >
> > We sincerely thank the reviewer for the follow-up feedback and for recognizing the improvements made in our initial rebuttal. We appreciate your continued engagement and your thoughtful suggestions for strengthening the paper further.
> >
> > ## Long-Context Evaluation
> >
> > We agree that ultra-long context evaluation is critical and can be sensitive to attention. Our current results demonstrate that our method scales well with sequence length up to 8K, and shows minimal accuracy degradation even at 50% sparsity. Our method preserves all tokens and maintains a full KV cache, which differentiates it from token-pruning strategies that often degrade under such conditions. Crucially, our method retains and activates the most imporatant heads which do compute the full attention algorithm, ensuring that strong signals are always propagated. As a result, we expect the approach to generalize well to longer context lengths while maintaining consistent performance and delivering throughput improvements of up to 2.2×.
> >
> > To strengthen our work and earn your confidence, **we have already begun experiments on InfiniteBench and will share preliminary results on at least one of the models during the discussion period**. This is a priority for us. We also commit to including comprehensive long-context results in the revised manuscript.
> >
> > ## Stronger models
> >
> > We agree that demonstrating robustness on the largest and most capable models, such as Qwen3 235B, is important. We want to assure the reviewer that we are fully committed to this evaluation. Given the significant computational resources required to run experiments on a 235B+ scale model, our immediate focus for the discussion period is on delivering the InfiniteBench results you requested. However, we are actively securing the necessary resources and commit to including an evaluation on at least one model at the >200B scale in the camera-ready version of the paper.
> >
> > We are confident that these planned experiments will fully address your remaining concerns and solidify the contributions of our paper. Thank you again for helping us improve the quality of our work.

---

> > > ### Author Response · Authors · 2025-08-09
> > > **Update on Long Context Evaluation**
> > >
> > > As part of our ongoing commitment to address your feedback, we have begun running ultra-long context experiments (up to 120k tokens) on InfiniteBench. These evaluations are significantly more time-intensive than shorter-context benchmarks, as each subtask requires ~8 hours with chunked generation on an A100 GPU to avoid out-of-memory errors. Despite the limited time, we have completed three representative tasks for both the dense and sparse (0.625) Llama-3.1-8B-Instruct models:
> > >
> > > | InfiniteBench tasks   | Average context lengths | Llama-3.1-8b-dense | Llama-3.1-8b-sparse-0.625 |
> > > |-----------------------|-------------------------|--------------------|--------------------------|
> > > | passkey               | 122.4k | 99.8% | 99.9% |
> > > | longdialogue_qa_eng   | 104k | 11% | 9% |
> > > | math_find             | 88k | 26% | 24% |
> > >
> > > For passkey (similar to needle-in-a-haystack retrieval), the sparse model slightly outperformed the dense baseline. We hypothesize that this is due to our top-k head selection preserving only the strongest attention signals, which may help focus retrieval in extremely long contexts. On other tasks, we observed the same graceful degradation trend seen in shorter-context settings.
> > >
> > > We will continue these evaluations and include the full InfiniteBench results covering more tasks and multiple model families in the revised manuscript.
> > >
> > > Thank you again for your constructive and insightful review. We sincerely hope that these new findings will merit a positive reconsideration of your assessment.

---

### Official Review · Reviewer_sVp2 · 2025-07-03

**Clarity:** 3
**Significance:** 3
**Originality:** 2
**Rating:** 4
**Confidence:** 3

**Summary:**

LLMs have been known to have very sparse activations. With specific choices of rectifying activations in MLP layers, such as ReLU, it has been observed that the activations are sparse. Similarly, attention heads have been shown to sparsely activate in the sense of their contributions to the output representation. The costs of these layers during inference depend on the batch size and length of the sequences involved. For short sequences in unbatched settings, the MLP layers tend to be the bottleneck, with runtime dominated by the time to read weights from memory. For longer sequences, the cost of reading the KV cache in the attention layers dominates the inference time. On the other hand, batching makes the MLP layers less sparse as the union of activations over the batch samples slowly approaches the dense computation. Furthermore, the MLP layers are memory-bound until the batch size hits a sufficient size, and increasing the batch size doesn't add to the cost. However, attention layers still maintain sparsity since the KV caches are independent for each batch item. The authors thoughtfully chose to optimize inference for the long-sequence and large-batch-size regime, where attention is the bottleneck. They do so by developing GPU kernels to exploit head sparsity effectively.

**Questions:**

- Is "head sparsity" a standard term? I was a bit confused initially about what it referred to. It might help define it early on in the introduction/abstract.
- The head sparsity remains stable and batch-invariant only for independent batch samples. What about use cases where we wish to generate multiple samples for the same batch item? The verifier in speculative decoding, autocomplete usecases, beam search, etc. sample multiple tokens from the same sequence.
- The paper claims decode stage dominates end-to-end latency, especially when generating long outputs spanning hundreds or thousands of tokens. I don't think this is necessarily true because the prefill stage has a quadratic computational complexity O(L^2) where L is the prefix length, and the decode stage has linear complexity O(L * D) where D is the number of decode steps and D << L.
- The paper claims to be the *first* to demonstrate that contextual sparsity scales with batch size. I'd argue that this has been a well-known fact, and most works aimed towards long-context efficiency would continue to work in batched settings. It's a soft opinion and is very subjective, but I'm curious to know if the authors have looked into works focusing on long-context efficiency.
- Are there any dead neurons in the initial layers? Fig 1 (b) seems to be suggestive of dead neurons and could be suitable for pruning.
- What are the percentages in Fig 2 (a)? What layer(s) are being plotted? It can be guessed from the text in 3.2, but I think the figure caption should be self-standing.
- What is the loss function used to train the routers?
- What does accuracy refer to in Figure 4?
- Have the naive implementations of the gather + attention operations been tested with different compilers? The operations sound like they can be fused easily by a compiler. I am not sure if the publicly available versions of the compilers support fusing arbitrary operations with custom calls yet (and I guess flash attention would require a custom call), but I don't think flash attention would help with performance until you hit a certain batch size.

**Ethical Concerns:**

["NO or VERY MINOR ethics concerns only"]

**Final Justification:**

The authors have identified and executed the idea well. There are some concerns with loss in performance on certain downstream tasks, but their method might still be Pareto optimal (which is not established in the paper but probably is the case). The new kernel as a contribution doesn't feel sophisticated enough to warrant a publication, but it's possible that I missed a few details; the authors do not disclose the code nor did they document enough implementation details highlighting the complexity. I'm retaining my score at 4, but would give 4.5. I'd have increased to 5 if the implementation is technically complex. Overall, the concern is whether this should have been a bug report to the compiler's repository or a paper. It would have been interesting to see results on other compilers such as XLA.

**Limitations:**

Yes, mostly. I think this method wouldn't scale for multiple independent samples for the same sequence (like in beam search or the verifier in speculative decoding).

**Paper Formatting Concerns:**

Nil.

**Quality:**

3

**Strengths And Weaknesses:**

I'm leaning towards increasing the score if the authors make good-faith effort to address the weaknesses below and answer the questions raised.

**Strengths:**
1. The paper overall is a complete piece of work and is written well.
2. The evaluations are thorough and cover a very broad set of models.
3. Impactful work.

**Weakness:**
1. No description of the loss functions used to train the routers in the main text.
2. No error estimates for results (Fig 4 and Table 1). The standard error for a lot of downstream evaluations are in several percentage points and is therefore extremely important to report error bars.
3. In general, downstream evals tend to be noisy and statistically unreliable. On the other hand, training metrics such as perplexity/top1-accuracy tend to be statistically reliable, but often poorly correlate with downstream performance. However, reporting results on both sets of metrics may improve the fidelity of the results. It's important since even a 1% gap in training top1 accuracy at scales of 70B can be as costly as doubling the model size, given the diminishing returns of the scaling laws. \*
4. There should be evaluations on long-form text generation. Is it possible that the approximation holds good most of the time, but the model fails catastrophically occasionally? But such occasional 1 in 100 failures can completely break decoding in long-form generation. The averages in the evaluations, most of them short-form generation, could easily hide these small modes.

\* Since this work is primarily approximating a pretrained model, I suspect that good retention on training metrics could likely correlate with good retention on downstream performance, as approximations don't fundamentally alter the model to cause dramatic changes in downstream evaluations. I am not sure; this is just a guess, and I might be wrong.

**Presentation Issues:**
1. Please ensure that the figure captions can stand on their own. What are the percentages in Fig 2 (a)?

**Minor nitpicks and soft opinions:**
1. I found it quite hard to understand the list of contributions at the end of the introduction. There are too many new terms and pharses being introduced without context, and they are not likely to be understandable without reading the entire paper (which defeats its purpose). Example terms: "Selective" GEMMs, "layer-wise top-k optimization strategy", "Group sparsity".
2. The list of contributions uses the phrase "negligible accuracy loss". It would be useful to report the exact top1 accuracy loss to make it clear; "negligible" is very subjective. I'd be concerned with even a 0.5% loss for very large models because that's the gap you see with doubling the model size at those scales.
3. There could be very specific failure modes with this kind of approximation, like the one highlighted in weakness comment 4. Would it be possible to construct cases where this approximation fails? The fact that some downstream evals seem to be impacted more than others suggests that there could be specific instances where this approximation fails.
4. I think the following excerpt from the paper is just one interpretation of the results. It could also be the case that most downstream don't catch issues. What is common in the evals that do worse than others?
> This task-dependent sensitivity is particularly encouraging, as it suggests that most tasks can be served with only the most critical heads, while harder tasks could potentially activate more for higher accuracy within the same batch.
5. Table 2 doesn't add value to the paper if the generation throughput or some other runtime performance quantity is not reported. It's possible to trade throughput for quality; hence, reporting just quality isn't useful.

---

> ### Author Rebuttal · Authors · 2025-07-31
>
> We sincerely thank the reviewer for their constructive feedback. These points have helped us improve the paper's rigor and clarity. We have taken action on all suggestions as detailed below.
>
> ---
> ## **Responses to Weaknesses**
>
> ## **1. Router Loss Function**
> We agree this information is better placed in the main text. Details on our router training, including the use of Binary Cross-Entropy (BCE) loss, were in Appendix C.
>
> ## **2. Error Estimates for Results**
> This is an excellent suggestion. We will update all figures and tables with error bars in the revised version to ensure statistical reliability.
>
> ## **3. Inclusion of Perplexity/Top-1 Accuracy Metrics**
> We agree that including training metrics would strengthen the evaluation. In Section 3 and Figure 2(a), we used the relative change in perplexity as a guiding signal to study head sparsity. We will report these training metrics for all evaluated models in the revised manuscript to ensure a more complete analysis.
>
> ## **4. Evaluation on Long-Form Generation**
> This is a critical point. We have run extensive new experiments on LongBench-e with three instruction-tuned models. The results confirm your intuition: our high-fidelity approximation leads to robust downstream performance without catastrophic failures. Our sparse models show only a minor, graceful degradation. Accuracy reported out of 100. We have also included additional results in the response to reviewer nrZF for MMLU_PRO.
>
> | Metric                | Llama-3.1-8B-Inst_dense | Llama-3.1-8B-Inst_sparse_0.5 | Llama-3.1-8B-Inst_sparse_0.625 | Mistral-7B-Inst_dense | Mistral-7B-Inst_sparse_0.5 | Qwen2.5-14B-Inst_dense | Qwen2.5-14B-Inst_sparse_0.5 |
> |---|---|---|---|---|---|---|---|
> | 2wikimqa              | 15.28 | 14.28 | 14.50 | 32.30 | 31.99 | 13.76 | 12.52 |
> | gov_report            | 34.66 | 31.79 | 33.56 | 28.64 | 28.60 | 30.99 | 27.34 |
> | hotpotqa              | 17.50 | 15.17 | 15.46 | 38.71 | 37.11 | 14.08 | 14.35 |
> | lcc                   | 69.56 | 60.72 | 63.70 | 56.78 | 56.11 | 67.90 | 60.41 |
> | multi_news            | 25.45 | 25.08 | 25.56 | 22.60 | 22.18 | 21.53 | 21.70 |
> | multifieldqa_en       | 27.79 | 27.92 | 26.61 | 32.91 | 32.09 | 35.18 | 32.12 |
> | passage_count         | 15.75 | 18.22 | 18.68 | 4.04 | 4.84| 7.64 | 11.05 |
> | passage_retrieval_en  | 97.12 | 94.24 | 95.46 | 31.67 | 37.33 | 89.64 | 94.61 |
> | qasper                | 11.83 | 11.85 | 12.49 | 30.04 | 28.34 | 10.67 | 10.94 |
> | repobench-p           | 55.31 | 45.19 | 48.76 | 47.14 | 41.77 | 47.62 | 40.15 |
> | samsum                | 42.73 | 38.39 | 40.79 | 40.84 | 39.74 | 44.69 | 41.01 |
> | trec                  | 71.00 | 71.67 | 71.67 | 61.33 | 61.33 | 75.00 | 75.00 |
> | triviaqa              | 92.81 | 89.72 | 90.72 | 83.11 | 82.68 | 88.43 | 86.61 |
> | LongBench-e average   | 44.37 | 41.86 | 42.92 | 39.24 | 38.78 | 42.09 | 40.60 |
>
> | Model  / LongBench-e accuracy by context length | 0-4k | 4-8k | 8k+ |
> |---|---|---|---|
> | Llama-3.1-8B-Inst_dense              | 44.01 | 44.30 | 44.80 |
> | Llama-3.1-8B-Inst_sparse-0.5         | 42.45 | 42.13 | 41.02 |
> | Llama-3.1-8B-Inst_sparse-0.625       | 43.35 | 43.00 | 42.41 |
> | Mistral-7B-Inst_dense                | 44.26 | 39.09 | 34.37 |
> | Mistral-7B-Inst_sparse_0.5           | 44.39 | 38.34 | 33.60 |
> | Qwen2.5-14B-Inst_dense               | 42.45 | 41.51 | 42.31 |
> | Qwen2.5-14B-Inst_sparse              | 40.46 | 40.47 | 40.87 |
>
> Across all these new, intensive evaluations, we observed no decoding collapses. This provides strong empirical evidence of stability for long-form generation. We will add a summary of these new results to the paper and are happy to include further analysis in the appendix.
>
> ---
>
> ## **Addressing Presentation Issues**
>
> We will revise all the captions to be self-contained. The percentages in Figure 2(a) indicate the relative increase in validation perplexity over the dense baseline, as detailed in Section 3.2.
>
> ## **Addressing Soft Opinions**
>
> - **Introduction Clarity, "Negligible Accuracy Loss":**  In the revision, we will rewrite the introduction to be more self-contained, briefly defining key terms as they are introduced. We will also report precise accuracy instead of using subjective terms to make our claims more accurate.
>
> - **Constructing Failure Cases:**  It is certainly possible to construct cases where our approximation is less effective, though this would require significant additional work. We see this as an exciting continuation of our research. Being able to identify such cases would allow for a dynamic, query-wise head activation strategy, paving the way for potentially lossless sparse inference.
>
> - **Analysis of Task Sensitivity:** While we have not observed catastrophic failures, we have noticed that harder reasoning, code, math benchmarks (e.g., MMLU-PRO Math, GSM8K, lcc, repobench) are more susceptible to accuracy degradation than other tasks. This aligns with the idea that more active heads perform better for harder queries.
>
> - **Value of Table 2:** This is a very helpful suggestion. We will include generation throughput in our revision, covering both single-query and large-batch scenarios to provide a more complete evaluation of the proposed approaches.
> ---
>
> ## **Answers to Questions**
>
> ### **Q: Is "head sparsity" a standard term?**
> **A:** Head sparsity isn't yet a standard term. We will define it early on to make the idea clear.
>
>
> ### **Q: Does batch-invariance hold for use cases like beam search or speculative decoding?**
> **A:**
> In standard beam search, a beam width of k is handled by expanding the batch dimension, treating each beam as an independent sequence. Therefore, our batch-invariance property holds, as each parallel beam can activate a different set of heads. However, for a highly optimized implementation where beams can share the KV cache and IO for the shared prefix, the head sparsity can potentially decrease depending on the overlap of head activations.
>
> SpecDecode: The drafting stage can still take advantage of our approach and we should be able to make batched speculative drafting more efficient. The verification stage can potentially have decreased head sparsity as different verification tokens within the batch can activate different heads. Exploring these topics would be an exciting future line of work.
>
> ### **Q: Is the decode stage always the bottleneck over prefill?**
> **A:** You are correct about the prefill cost in long-prompt, short-generation scenarios (D≪L). Our work focuses on use cases where the decode stage is the practical bottleneck, such as long-form generation (D≥L) and multi-turn chat. In these applications, the decode stage's sequential, memory-bound nature dominates user-perceived latency, especially since the initial prefill cost is an amortized, one-time expense. From a systems perspective, while prefill has quadratic complexity, it is highly parallel and often achieves near-full GPU utilization. In contrast, the decode stage is inherently sequential and memory-bandwidth bound, making its latency the practical bottleneck in these use cases. We recognize our original claim lacked this precision and will revise the paper to clarify this specific context.
>
> ### **Q: Is this the "first" work to show contextual sparsity scales with batch size?**
> **A:** Our claim was made in the specific context of contextual sparsity; existing methods often report reduced speedup with larger batches. In Appendix A.1, we survey recent contextual sparsity approaches and document this behavior. In Appendix A.2 we discuss token level sparsity techniques that skip or prune tokens for improved long context efficiency. We believe combining those methods with Polar Sparsity could yield even greater throughput gains. We agree with your general point that most attention optimization should work well in batched settings. Our key contribution is being the **first to design, build, and demonstrate practical, high-performance GPU kernels (Selective FlashAttention) that convert theoretical *attention head sparsity* into measured, wall-clock speedups in high-throughput scenarios.**
>
> ### **Q: Are there dead neurons in the initial layers?**
> **A:** This is a sharp observation. As we discuss in Appendix B.1, these are not 'dead' neurons but rather a visualization of extremely high contextual sparsity, where very few neurons activate for any single query.
>
> ### **Q: What does "accuracy" refer to in Figure 4?**
> **A:** The 'accuracy' in Figure 4 refers to the zero-shot accuracy on downstream tasks. We will revise the caption and make sure all our graphs are self-sufficient.
>
> ### **Q: Have you tested against advanced compilers that could fuse these operations?**
> **A:** We tested the naive implementation with torch.compile, and while it helps, it still doesn't perform better than the standard dense baseline.
>
> batch size 64, 72 Heads, seq len 2k, MHA, 50% sparsity, A5000 GPU
> | method | latency (ms) |
> |---|---|
> | naive gather + flashattn | 14.7 |
> | torch.compile(gather + flashattn) | 9.8 |
> | dense flashattn | 8.7 |
> | select flashattn (ours) | 4.8 |
>
> ---
>
> ## **Conclusion**
>
> We'd like to conclude by expressing our genuine appreciation for your review. We are truly grateful for a review that was not only technically rigorous but also deeply engaged with the core value of our work. Your feedback has been pivotal in helping us clarify our contributions and better articulate the paper's impact. We believe the manuscript is significantly stronger now and hope we have fully addressed your points to merit an increased score.

---

> ### Comment · Reviewer_sVp2 · 2025-08-03
>
> Thank you for your response. I'm satisfied with most of the rebuttal, and have follow-ups for the remaining parts.
>
> Response to **2. Error Estimates for Results**
>
> I think many frameworks such as eval-harness report the standard errors along with the performance numbers. If not, it's fairly easy to obtain CLT-based estimate using minibatches. Would it be possible to obtain error bars for the long-form evals for the tables in section 4 of the rebuttal?
>
> Response to **Q: Does batch-invariance hold for use cases like beam search or speculative decoding?**
>
> The authors have clarified the question on batch-invariance property in certain beam search settings (e.g., long shared prefix) and verification step in speculative decoding. As far as my memory goes, the authors did not note it explicitly in their paper. I'd expect the authors to include it in their revised version.
>
> Response to **4. Evaluation on Long-Form Generation**
>
> Would it be possible to manually look for differences in decoded tokens for the baseline and the sparse versions? To me it looks like the **performance of the sparse models is degrading with length faster than the baseline** for Llama-3.1-8B. It's less dramtic for Mistral and Qwen looks alright. However, it's difficult to draw definitive conclusions without having an idea of the standard errors for these measurements. The numbers for Qwen baseline seems to suggest std errors to be substantial.
>
> Response to **Q: Have you tested against advanced compilers that could fuse these operations?**
>
> Would it be possible to check the performance with naive attention? I am not entirely convinced that FlashAttention offers any advantage at small batch sizes. I'm not deeply familiar with the PyTorch ecosystem and assume that backend used was torch inductor, but I'd imagine that naive attention would spill all the operations into the computational graph, which might allow the compiler do a better job with fusions.

---

> > ### Author Response · Authors · 2025-08-05
> > **Response to Reviewer sVp2 (Part 1/2)**
> >
> > # Response to comments and questions (Part 1/2)
> >
> > Thank you for your continued engagement with our work and for your insightful follow-up questions. We appreciate the opportunity to provide further clarification and are pleased that our initial rebuttal was satisfactory. We provide the additional data and analysis you requested below.
> >
> > ## Error Estimated for LongBench Eval
> >
> > We have now calculated the standard error for our LongBench results, and the updated tables, including error bars, are provided below.
> >
> > | Metric | Llama-3.1-8B-Inst_dense | Llama-3.1-8B-Inst_sparse_0.5 | Llama-3.1-8B-Inst_sparse_0.625 | Mistral-7B-Inst_dense | Mistral-7B-Inst_sparse_0.5 | Qwen2.5-14B-Inst_dense | Qwen2.5-14B-Inst_sparse_0.5 |
> > |---|---|---|---|---|---|---|---|
> > | 2wikimqa              | 15.28 ± 0.86 | 14.28 ± 0.80 | 14.50 ± 0.79 | 32.30 ± 2.53 | 31.99 ± 2.53 | 13.76 ± 0.82 | 12.52 ± 0.75 |
> > | gov_report            | 34.66 ± 0.33 | 31.79 ± 0.33 | 33.56 ± 0.34 | 28.64 ± 0.44 | 28.60 ± 0.44 | 30.99 ± 0.33| 27.34 ± 0.29 |
> > | hotpotqa              | 17.50 ± 0.86 | 15.17 ± 0.64 | 15.46 ± 0.75 | 38.71 ± 2.53 | 37.11 ± 2.48 | 14.08 ± 0.75 | 14.35 ± 0.81 |
> > | lcc                   | 69.56 ± 2.44 | 60.72 ± 1.73 | 63.70 ± 1.73 | 56.78 ± 1.84 | 56.11 ± 1.90 | 67.90 ± 1.74 | 60.41 ± 1.82 |
> > | multi_news            | 25.45 ± 0.34 | 25.08 ± 0.32 | 25.56 ± 0.31 | 22.60 ± 0.44 | 22.18 ± 0.42 | 21.53 ± 0.27 | 21.70 ± 0.27 |
> > | multifieldqa_en       | 27.79 ± 1.78 | 27.92 ± 1.66 | 26.61 ± 1.63 | 32.91 ± 2.75 | 32.09 ± 2.76 | 35.18 ± 2.19 | 32.12 ± 1.87 |
> > | passage_count         | 15.75 ± 1.88 | 18.22 ± 2.02 | 18.68 ± 2.08 | 4.04 ± 0.99 | 4.84 ± 1.04 | 7.64 ± 0.93| 11.05 ± 1.31 |
> > | passage_retrieval_en  | 97.12 ± 0.75 | 94.24 ± 0.98 | 95.46 ± 0.90 | 31.67 ± 2.69 | 37.33 ± 2.80 | 89.64 ± 1.27 | 94.61 ± 1.01 |
> > | qasper                | 11.83 ± 0.87 | 11.85 ± 0.79 | 12.49 ± 0.90 | 30.04 ± 2.29| 28.34 ± 2.23 | 10.67 ± 0.9 | 10.94 ± 0.81 |
> > | repobench-p           | 55.31 ± 2.10 | 45.19 ± 1.66 | 48.76 ± 1.76 | 47.14 ± 1.80 | 41.77 ± 1.84 | 47.62 ± 2.15 | 40.15 ± 2.06 |
> > | samsum                | 42.73 ± 0.96 | 38.39 ± 0.93 | 40.79 ± 0.96 | 40.84 ± 0.88 | 39.74 ± 0.83 | 44.69 ± 0.87 | 41.01 ± 0.87 |
> > | trec                  | 71.00 ± 2.62 | 71.67 ± 2.61 | 71.67 ± 2.61 | 61.33 ± 2.82 | 61.33 ± 2.82 | 75.00 ± 2.5 | 75.00 ± 2.5 |
> > | triviaqa              | 92.81 ± 1.62 | 89.72 ± 1.50 | 90.72 ± 1.43 | 83.11 ± 1.97| 82.68 ± 2.0 | 88.43 ± 1.62 | 86.61 ± 1.69 |
> > | LongBench-e average   | 44.37 ± 8.23 | 41.86 ± 7.91 | 42.92 ± 8.01 | 39.24 ± 5.40 | 38.78 ± 5.30 | 42.09 ± 8.21 | 40.60 ± 8.18 |
> >
> > | Model  / LongBench-e | 0-4k | 4-8k | 8k+ |
> > |---|---|---|---|
> > | Llama-3.1-8B-Inst_dense | 44.01 ± 7.7 | 44.30 ± 8.64 | 44.80 ± 8.73 |
> > | Llama-3.1-8B-Inst_sparse-0.5 | 42.45 ± 7.45| 42.13 ± 8.17 | 41.02 ± 8.28 |
> > | Llama-3.1-8B-Inst_sparse-0.625 | 43.35 ± 7.7 | 43.00 ± 8.14 | 42.41 ± 8.37 |
> > | Mistral-7B-Inst_dense | 44.26 ± 4.93 | 39.09 ± 5.52 | 34.37 ± 6.29 |
> > | Mistral-7B-Inst_sparse_0.5 | 44.39 ± 4.98 | 38.34 ± 5.43 | 33.60 ± 6.21 |
> > | Qwen2.5-14B-Inst_dense | 42.45 ± 7.79 | 41.51 ± 8.34 | 42.31 ± 8.61 |
> > | Qwen2.5-14B-Inst_sparse_0.5 | 40.46 ± 7.88 | 40.47 ± 8.28 | 40.87 ± 8.47 |
> >
> > We will include these updated tables in the final version of the paper.
> >
> > ## Spec Decode, Beam Search
> >
> > We will add a detailed discussion of these use cases, addressing beam search and speculative decoding, in the limitation and future work section of our revised manuscript.
> >
> > (Additional response included in the next official comment)

---

> > > ### Author Response · Authors · 2025-08-05
> > > **Response to Reviewer sVp2 (Part 2/2)**
> > >
> > > # Response to comments and questions (Part 2/2)
> > >
> > > ## LongBench Manual Evaluation
> > >
> > > Following your suggestion, we manually compared generated outputs for Llama-3.1-8B-Inst_dense and Llama-3.1-8B-Inst_sparse_0.625. Our key findings are:
> > >
> > > + **No Catastrophic Failures**: We found no instances of catastrophic failure or generation collapse. The sparse model consistently produces coherent, relevant, and high-quality text, even on the longest contexts.
> > >
> > > + **Qualitative Similarity**: The generated text from both models is qualitatively very similar even at longer context lengths. The differences are largely stylistic (e.g., phrasing, word choice), as illustrated by the samsum (Summarization) examples below. For classification tasks like trec, the outputs were often identical.
> > >
> > > + **Slightly Higher Verbosity**: We noted that the sparse Llama-3.1-8B model generated slightly more verbose text. Since most LongBench tasks use ROUGE scores, which penalize length mismatches, this verbosity offers a direct explanation for the slight decrease in scores, as the semantic content remained equivalent to the dense model's output.
> > >
> > > | Example |Generation mode | Text generated | Rouge score |
> > > |---|---|---|---|
> > > | 1 |Dense | "Mia and Steven are going to meet at 8 pm to have Chinese food before the movie at 9 pm. ...." | 0.428 |
> > > | |Sparse | "Mia and Steven are going to the movies at 9 pm. They will meet at 8 pm to grab something to eat at a Chinese restaurant. ...." | 0.375 |
> > > | |Ground truth | "Mia and Steven want to grab some Chinese at 8." | - |
> > > |2 | Dense | "Pam lost her iPhone. She thinks she left it on the counter when she was paying. She talked to all the stores she visited but no one found it. Dot suggests that Pam can display a message on the screen to help find it. ...."  | 0.408 |
> > > | | Sparse | " Pam lost her iPhone. She tried to find it in the stores she visited, but it was nowhere to be found. Dot suggests that Pam log into her Apple account to display a message on the screen to help locate the phone. ...." | 0.370 |
> > > | | Ground truth | "Pam has probably left her phone when she was shopping. Dot suggests there is a way to get it back." | |
> > >
> > > This qualitative result supports our hypothesis that by preserving the top-k most influential heads, the strongest signals are always propagated and the model retains its core reasoning and generation capabilities. We will include this analysis in the revised manuscript.
> > >
> > > ## Compilers for operation fusion
> > >
> > > This is an excellent suggestion. As you rightly suspected, for a batch size of 1, a compiled naive PyTorch attention implementation is indeed the fastest approach, as torch.compile effectively fuses the exposed operations.
> > >
> > > However, our new experiments show this advantage is limited to that specific low-concurrency case. As the table below illustrates, the performance of the compiled naive attention degrades rapidly with batch size. By batch size 4, it is already significantly slower than the dense FlashAttention baseline, and at batch size 64, it is over 4x slower than our optimized kernel.
> > >
> > > Since our work's primary contribution is **accelerating high-throughput, large-batch inference**, our SelectAttn kernel demonstrates a significant and practical speedup where it matters most, affirming its superior scaling and value.
> > >
> > > **Details:** MHA, 72 heads, Seq len 2000, head dim 128, 50% head activation, time in ms, NVIDIA A5000 GPU
> > >
> > > | Attention Implementation / Batch Size | 1     | 2     | 4     | 8     | 16    | 32    | 64    |
> > > |--------------------------------------|-------|-------|-------|-------|-------|-------|-------|
> > > | Dense FlashAttn                      | 0.316 | 0.435 | 0.596 | 1.15  | 2.25  | 4.46  | 8.82  |
> > > | SelectAttn                           | 0.25  | 0.252 | 0.312 | 0.642 | 1.16  | 2.26  | 4.51  |
> > > | torch.compile(gather + naïve Pytorch attn) | 0.185 | 0.534 | 1.035 | 2.04  | 4.23  | 8.77  | 19.25 |
> > > | Gather + FlashAttn                    | 0.240 | 0.448 | 0.867 | 1.73 | 3.57 | 7.04 | 14.7 |
> > >
> > >
> > > ## Final Remarks
> > >
> > > We thank you once again for your meticulous review and constructive dialogue. Your feedback has been invaluable in helping us strengthen the paper. We hope these comprehensive responses and new results have fully addressed your concerns and will merit a higher rating for our work.

---

> > > > ### Comment · Reviewer_sVp2 · 2025-08-08
> > > >
> > > > Is there a theoretical estimate of the expected runtime of the kernel?

---

> > > > > ### Author Response · Authors · 2025-08-08
> > > > > **Theoretical kernel runtime**
> > > > >
> > > > > ## **Theoretical estimate of kernel runtime**
> > > > >
> > > > > We can model the expected runtime of our kernel using a memory-bandwidth-bound roofline model, which is appropriate for the decode stage where attention performance is dominated by reading the large Key-Value (KV) cache from HBM. The table below defines the symbols used in our analysis.
> > > > >
> > > > > | Symbol               | Meaning                                     |
> > > > > | -------------------- | ------------------------------------------- |
> > > > > | $B$                  | Batch size                                  |
> > > > > | $H$                  | Total attention heads                       |
> > > > > | $K$                  | Active heads                                |
> > > > > | $\rho$               | Head density, $K/H$                       |
> > > > > | $L$                  | Context length (tokens already in KV cache) |
> > > > > | $d$                  | Head dimension                              |
> > > > > | $s$                  | Bytes per element (e.g., bf16: $s=2$)       |
> > > > > | $\Phi_{\text{eff}}$  | Effective FLOP/s                            |
> > > > > | $BW_{\text{eff}}$ | Effective memory bandwidth (bytes/s)        |
> > > > >
> > > > > **Theoretical expected runtime** =
> > > > > $
> > > > > T \approx \max\left( \frac{\text{TotalFlops}}{\Phi_{\text{eff}}}, \frac{\text{TotalMemIO}}{BW_{\text{eff}}} \right) \approx \frac{\text{TotalMemIO}}{BW_{\text{eff}}}
> > > > > $
> > > > >
> > > > > For the dense baseline, the kernel reads the entire KV cache from HBM for all attention heads. In contrast, our kernel reads only the k active heads from the KV cache, plus a small amount of data for the indices that specify which heads to load.
> > > > >
> > > > > + IO for dense flash attention: $IO_{\text{dense}} \approx KV_{\text{reads}} = 2\  \times B\ \times H\ \times L\ \times d \times s\$
> > > > > + IO for Select flash attention:$IO_{\text{select}} \approx KV_{\text{reads}} + Index_{\text{reads}}  =2\  \times B\ \times K\ \times L\ \times d \times s\ + B\ \times K\ \times s\$
> > > > >
> > > > > We can compute the estimated theoretical runtime based on the IO and memory bandwidth.
> > > > >
> > > > > **Theoretical Runtime (Dense FlashAttention):**
> > > > > $
> > > > > T_{\text{dense}} \approx \frac{IO_{\text{dense}}}{BW_{\text{eff}}} = \frac{2 \times B \times H \times L \times d \times s}{BW_{\text{eff}}}
> > > > > $
> > > > >
> > > > > **Theoretical Runtime (Selective FlashAttention):**
> > > > > $
> > > > > T_{\text{select}} \approx \frac{IO_{\text{select}}}{BW_{\text{eff}}} = \frac{2 \times B \times K \times L \times d \times s + B \times K \times s}{BW_{\text{eff}}}
> > > > > $
> > > > >
> > > > > $\text{Speedup}_{\text{select}} \approx \frac{2 H L d}{2 K L d + K} $
> > > > >
> > > > > As batch size and sequence length scale up, the cost of reading index tensors in selective attention becomes negligible compared to the total KV memory access. This simplifies the theoretical speedup expression:
> > > > >
> > > > > $\text{Speedup}_{\text{select}} \approx \frac{2 H L d}{2 K L d + K}
> > > > > \longrightarrow \frac{H}{K} = \frac{1}{\rho} \quad \text{as } L, B \to \infty$
> > > > >
> > > > > This implies that, in the large-batch, memory-bound regime, the maximum achievable speedup approaches the ratio of total heads to active heads. For example, with H=72, K=36, the expected asymptotic speedup is ≈2×. This matches closely with our empirical speedup of 1.91× observed on H100 GPUs (as noted in our response to Reviewer bAJa). We will include this theoretical analysis in the revised version of the paper.

---

> ### Comment · Reviewer_sVp2 · 2025-08-09
>
> 1. Would it be possible to debug and find where exactly the compiler is failing for small batch sizes?
> 2. Is there something really non-trivial in your implementation that a good compiler cannot figure out on its own?

---

> > ### Author Response · Authors · 2025-08-09
> >
> > ## torch.compile in smaller batch sizes
> >
> > From a systems perspective, the performance difference can be explained by the materialization of an intermediate tensor in global memory, which our custom kernel is specifically designed to avoid. At batch size of 1, the gathered KV cache for a single layer is small enough to fit in the GPU's faster L2 cache and the compiler can effectively fuse the simple operations into a single graph that runs efficiently.
> >
> > However, as the batch size increases, the workload becomes increasingly memory-bandwidth bound and the compiler's approach breaks down. The sequence of operations gather + attention, even when compiled, almost certainly leads to the following execution pattern:
> > + Kernel 1 (Gather): Launches a kernel to gather the active heads from the full KV cache in HBM
> > + Writes a new, smaller intermediate selected KV cache tensor back into HBM
> > + Launches an attention kernel to read the intermediate tensor from HBM
> >
> > This extra HBM write/read roundtrip is the critical performance bottleneck that prevents the compiled naive implementation from scaling.
> >
> >
> > ## Non-Trivial contribition
> >
> > Our contribution is non-trivial because SelectAttention is a single, monolithic GPU kernel that avoids the intermediate tensor materialization described above. We did not simply chain a gather kernel and a FlashAttention kernel. Instead, we modified the souce code of FlashAttention algoirthm iteself such that each memory IO performs indexed, strided memory loads directly from the full KV cache in HBM into the on-chip SRAM. In essence, the "gather" operation is not a separate step; it is fused directly into each memory access pattern of the attention computation at the lowest level.
> >
> > Todays compilers cannot perform this optimizaiton for two primary reasons:
> > + Abstraction: Compilers operate on a higher level of abstraction and typically treat highly-optimized libraries like FlashAttention as opaque "custom calls." They cannot arbitrarily rewrite the internal CUDA logic of such a kernel.
> > + Complexity: The memory access patterns inside FlashAttention are already extremely complex and tuned for specific GPU hardware. Integrating indexed lookups while maintaining performance and avoiding bank conflicts is a significant manual engineering task.
> >
> > We are sincerely grateful for your engagement throughout this process. Your insightful questions have been invaluable in helping us to precisely articulate the technical novelty of our work, and the paper is significantly stronger as a result. We trust that our responses have fully addressed your questions.

---

### Official Review · Reviewer_nrZF · 2025-07-03

**Clarity:** 3
**Significance:** 2
**Originality:** 3
**Rating:** 4
**Confidence:** 4

**Summary:**

This paper proposes Polar Sparsity, a batch-friendly inference framework that combines a dynamic, layer‐wise top-k MLP router with a custom Selective GEMM kernel and a lightweight attention router with sparsity-aware Selective FlashAttention kernels to exploit high-throughput contextual sparsity. Built on top of FlashAttention and CUDA graphs, Polar Sparsity achieves throughput improvements across various models, demonstrating that contextual sparsity can be practically scaled to high-throughput LLM deployment with minimal changes.

**Questions:**

See Weaknesses.

**Ethical Concerns:**

["NO or VERY MINOR ethics concerns only"]

**Final Justification:**

The rebuttal has addressed most of my concerns. The proposed polar sparsity approach effectively resolves the batch inference issue associated with contextual sparsity in LLMs. While I acknowledge that polar sparsity leads to a relatively notable performance drop on math reasoning benchmarks, I recognize its contribution to enabling sparse batch inference. Therefore, I raise my score to weak acceptance.

**Limitations:**

yes

**Paper Formatting Concerns:**

No concerns.

**Quality:**

3

**Strengths And Weaknesses:**

Strengths:
1. The paper is well written and easy to understand.
2. The figures are well plotted to help understand the effectiveness of Polar Sparsity.

Weaknesses:
1. Under the single-query or small-batch setting—typical for interactive chat or RLHF—the speedup ratio is limited due to the extra routing overhead.
2. More experiments should be conducted on harder reasoning tasks (e.g., GSM-Hard, MATH).
3. Throughput performance could be tested on more devices, e.g., H100, consumer RTX, and AMD.
4. No performance comparisons to the methods deploying the batch inference with structured sparsity.
5. The training-based sparsification with customized sparse kernel design makes it hard to quickly deploy on a new model.

---

> ### Author Rebuttal · Authors · 2025-07-31
>
> We sincerely thank the reviewer for their time and for providing thoughtful and constructive feedback on our submission. We are encouraged that the reviewer found the paper well-written and the figures clear. We have carefully considered the weaknesses raised and have performed additional experiments and analysis to address them. We believe these additions significantly strengthen the paper and we hope the reviewer will reconsider their rating.
>
> Below, we address each of the weaknesses point-by-point.
>
> ## 1. Performance in Single-Query or Small-Batch Settings
>
> We appreciate the reviewer's observation regarding performance in low-batch settings. Our primary focus with Polar Sparsity is to maximize performance for high-throughput LLM serving, a dominant use-case in real-world cloud deployments. Most service providers leverage techniques like continuous batching to serve large numbers of concurrent requests (including chat models with RLHF), maximizing hardware utilization and cost-effectiveness [1]. Our work provides a novel solution to reduce latency and boost throughput in this critical, batch-heavy environment.
>
> That said, we want to emphasize that the overhead of our method is minimal.
>
> + Minimal Router Overhead: We have conducted extended router evaluation in Appendix C.1, where our ablation study shows that the attention routers are exceptionally lightweight and add negligible overhead.
>
> + Meaningful Low-Batch Speedups: Even in low-batch settings, Polar Sparsity delivers tangible performance gains. As shown in Figures 5 and 6 of our original submission, we achieve meaningful speedups, demonstrating the efficiency of our approach even when not operating at maximum throughput.
>
> While the most dramatic speedups are realized in high-throughput scenarios, our highly optimized routers and kernels ensure that the gains from sparsity consistently outweigh the minimal overhead, making our framework beneficial even in smaller-batch contexts.
>
> ## 2. Performance on Harder Reasoning Tasks
>
> This is an excellent suggestion. To demonstrate the robustness of Polar Sparsity on more complex tasks, we have conducted new experiments on the MMLU-PRO, GSM8K and LongBench benchmarks. These results, which we will incorporate into the revised manuscript, show that our sparse models maintain high accuracy with only a minor, graceful trade-off.
>
> Summary of New Benchmark Results for instruction tuned chat models:
>
> | Subject                    | Llama-3.1-8B-Inst_dense | Llama-3.1-8B-Inst_sparse_0.625 | Mistral-7B-Inst_dense | Mistral-7B-Inst_sparse_0.5 |
> |---|---|---|---|---|
> | MMLU_PRO_Biology           | 0.62 | 0.60 | 0.47 | 0.50 |
> | MMLU_PRO_Business          | 0.44 | 0.43 | 0.21 | 0.20 |
> | MMLU_PRO_Chemistry         | 0.26 | 0.24 | 0.12 | 0.11 |
> | MMLU_PRO_Computer Science  | 0.42 | 0.41 | 0.28 | 0.29 |
> | MMLU_PRO_Economics         | 0.53 | 0.51 | 0.34 | 0.35 |
> | MMLU_PRO_Engineering       | 0.24 | 0.27 | 0.16 | 0.15 |
> | MMLU_PRO_Health            | 0.51 | 0.49 | 0.31 | 0.30 |
> | MMLU_PRO_History           | 0.41 | 0.37 | 0.25 | 0.29 |
> | MMLU_PRO_Law               | 0.28 | 0.28 | 0.17 | 0.17 |
> | MMLU_PRO_Math              | 0.39 | 0.35 | 0.17 | 0.17 |
> | MMLU_PRO_Other             | 0.46 | 0.44 | 0.32 | 0.29 |
> | MMLU_PRO_Philosophy        | 0.43 | 0.42 | 0.26 | 0.24 |
> | MMLU_PRO_Physics           | 0.34 | 0.34 | 0.19 | 0.18 |
> | MMLU_PRO_Psychology        | 0.59 | 0.59 | 0.44 | 0.43 |
> | MMLU_PRO Overall Accuracy  | 0.41 | 0.40 | 0.25 | 0.24 |
> | GSM8K                      | 0.75 | 0.71 | 0.34 | 0.32 |
> | LongBench                  | 0.443 | 0.429 | 0.392 | 0.388 |
>
> These results confirm that Polar Sparsity preserves model capabilities on challenging reasoning and long-context tasks, with overall accuracy showing only a minor degradation (e.g ~1% on MMLU_PRO Overall Accuracy) relative to the dense models.
>
> Crucially, we want to highlight that this is the first work to demonstrate wall-clock speedups by enforcing sparsity on attention heads. The minor accuracy degradation opens an exciting avenue for future work toward completely lossless sparsity. As we explain in the paper, the batch-invariant nature of our head sparsity router means we can dynamically apply different sparsity levels to different queries within the same batch. This would allow "harder" queries to use more heads, preserving quality, while "easier" queries benefit from higher acceleration, a unique and powerful capability of our framework.
>
> ## 3. Throughput Performance on More Devices
>
> We agree that demonstrating performance across a range of hardware is vital. We have now benchmarked our sparsity-aware Selective FlashAttention kernel on a variety of modern GPUs, including the NVIDIA RTX A5000, A100, and H100.
>
> The results below show that our kernel's speedup is consistent and significant across different hardware tiers.
>
> Attention Kernel Latency (ms) and Speedup (Batch=64, SeqLen=2k, Num Heads = 72, MHA)
> | Head Activation | A5000 Speedup | A100 Speedup | H100 Speedup |
> | :---: | :---: | :---: | :---: |
> | 1.0 (Dense) | 1.00x | 1.00x | 1.00x |
> | 0.8 | 1.17x | 1.22x | 1.24x |
> | 0.6 | 1.52x | 1.57x | 1.62x |
> | 0.5 | 1.80x | 1.82x | 1.91x |
> | 0.3 | 3.18x | 2.76x | 3.08x |
> | 0.1 | 7.93x | 5.59x | 8.23x |
>
> The data confirms that our approach is not architecture-specific. At a typical 50% head activation, we achieve a ~1.8x-1.9x speedup across all tested GPUs. While the kernel was developed on A100, its excellent performance on both consumer and datacenter hardware is highly encouraging. We will include these kernel benchmarks in the paper and commit to providing full end-to-end throughput results for all models on these devices in the revised manuscript.
>
> ## 4. Comparison to Structured Sparsity Methods
>
> We thank the reviewer for this question and would like to clarify our positioning. We consider contextual sparsity to be an advanced, dynamic form of structured sparsity. We did compare Polar Sparsity with Deja Vu, a state-of-the-art method that applies structured sparsity to MLP and linear layers. Our results in Figure 5 show that **Polar Sparsity is up to 2x faster than Deja Vu**, marking a significant improvement.
>
> Our work is distinct from traditional structured pruning techniques [2] for two critical reasons:
>
> + Addressing the Right Bottleneck: Most structured sparsity methods focus only on MLP/linear layers. In large-batch, high-throughput inference, the attention mechanism is often the primary bottleneck. Our work is the first to introduce custom kernels that bring the benefits of structured sparsity directly to the attention computation, which is a key novelty and contribution.
>
> + Training Efficiency: Traditional methods often require full, computationally intensive sparse training to prune model weights. Our approach is far more practical, requiring only the training of a lightweight router.
>
> ## 5. Deployment on New Models
>
> We appreciate the reviewer’s concern about the practicality of our training-based method. We wish to clarify that our approach is designed for fast and easy deployment and does not require full model retraining or fine-tuning.
>
> The MLP and attention routers are very lightweight and are trained completely separately from the frozen, pre-trained model weights. As detailed in Appendix C and the provided supplementary code, this process is highly efficient. For example, training the attention router for each model takes less than an hour on two A5000 GPUs.
>
> This makes our framework significantly more agile and easier to apply to new models compared to methods that depend on extensive and costly retraining of the entire model.
>
> ## Conclusion
>
> In conclusion, we have demonstrated that Polar Sparsity is a practical and impactful framework that (1) excels in the critical high-throughput serving scenario, (2) maintains strong performance on hard reasoning tasks, (3) generalizes across different modern GPU architectures, (4) outperforms existing structured sparsity methods by tackling the attention bottleneck, and (5) is lightweight and easy to deploy.
>
> We are grateful for the reviewer's feedback, which has helped us improve the paper. We believe these new results and clarifications address the concerns raised and demonstrate the significance of our work. We respectfully hope the reviewer will consider increasing their score.
>
>
> References
>
> [1] Orca: A Distributed Serving System for Transformer-Based Generative Models.
>
> [2] Accelerating Sparse Deep Neural Networks. arXiv:2104.08378.

---

> > ### Author Response · Authors · 2025-08-06
> > **Follow-up Discussion**
> >
> > Dear Reviewer nrZF,
> >
> > Thank you again for your constructive review of our paper. Given the extended discussion period, we wanted to briefly follow up on our rebuttal.
> >
> > In response to your feedback, our rebuttal includes:
> >
> > + **Strong accuracy on harder tasks**, confirmed with new results on MMLU-PRO, GSM8K, LongBench.
> >
> > + **Consistent speedups across hardware**, now benchmarked on H100 and RTX A5000 GPUs.
> >
> > + Further clarifications on deployment practicality and our novelty over other methods.
> >
> > We would be grateful to know if these updates help resolve your concerns. We are available and eager to discuss any further questions you may have.

---

> > ### Comment · Reviewer_nrZF · 2025-08-06
> >
> > Thank you for the rebuttal. Some of my concerns have been resolved. Below are the remaining concerns:
> >
> > 1. Regarding the performance on harder reasoning tasks, your current results on these benchmarks use instruction-tuned models, yet the two baselines are not specifically fine-tuned for reasoning. Evaluating reasoning models designed for reasoning benchmarks (e.g., distilled DeepSeek-R1 variants) would provide a clearer demonstration of the effectiveness.
> >
> > 2. Regarding the comparison to structured sparsity methods, Figure 5 should include a breakdown showing how much each proposed component contributes to the overall speedup versus DejaVu. That will clarify the individual effects of each design choice.
> >
> > 3. Regarding the throughput performance, only the performance of the Selective FlashAttention kernel was reported on different devices? How about the sparse GEMM kernel designed for linear layers?

---

> > > ### Author Response · Authors · 2025-08-09
> > >
> > > ## Reasoning Models
> > >
> > > Thank you for the suggestion. We have started evaluating reasoning-specialized models like DeepSeek-R1-Distill-Llama-8B but found that standard tools like lm-eval-harness are not equipped for their long, chain-of-thought outputs, leading to inaccurate automated scoring. As a result, the current pipeline reports the base Llama-3-8B-Inst achieving a higher MMLU-PRO score than the distilled reasoning variant, despite the latter often reaching the correct answer in manual inspection. These models often produce extended multi-hundred-token reasoning chains before the final answer, and the framework’s default limits and parsing logic frequently miss the correct output. We are adapting the evaluation pipeline to handle long-form reasoning outputs, and aim to include these results in the final version of the paper.
> > >
> > > In the meantime, we wish to re-emphasize that our method’s effectiveness has been demonstrated across five diverse model families (OPT, Llama-2, Llama-3, Mistral, and Qwen) on a wide range of benchmarks. We believe this broad validation, combined with our core contribution as the first work to demonstrate scalable wall-clock speedups from contextual attention sparsity, strongly supports the significance of our paper.
> > >
> > > ## Speedup contribution
> > >
> > > Thank you for this question, as it allows us to clarify a critical aspect of our comparison. The DejaVu framework, in its original form, is designed for single-query inference and relies on sparse General Matrix-Vector (GEMV) operations. It does not natively support the batched inference workloads that are the focus of our work.
> > >
> > > To create a meaningful, state-of-the-art baseline for batched contextual sparsity, we had to **adapt** the DejaVu approach. Specifically, for the MLP layers, we replaced its single-query GEMV kernel with our own batch-friendly Selective GEMM kernel. Therefore, in the MLP computations for the experiment shown in Figure 5, both our Polar Sparsity framework and our DejaVu baseline are using the **identical** Selective GEMM kernel.
> > >
> > > This means the entire performance difference shown in Figure 5 comes from our novel contribution to the attention layers. Thus, the speedup of Polar Sparsity over the adapted DejaVu baseline is entirely attributable to our sparsity-aware Selective FlashAttention kernel. We will make this experimental setup explicitly clear in the final manuscript and will likely add another graph to highlight the contribution of each kernel.
> > >
> > >
> > > ## Selective GEMM
> > >
> > > You've raised an excellent point. We initially focused our rebuttal on the attention kernel, as accelerating attention provides the most significant speedup in the large-batch scenarios targeted by our work. To provide a complete picture, we have benchmarked our Selective GEMM kernel across the A5000, A100, and H100 GPUs. The results show strong, consistent speedups that align with those of our attention kernel.
> > >
> > > Table: Selective GEMM Kernel Latency (ms) and Speedup vs. Dense cuBLAS
> > > (Batch=64, hidden_dim = 9216, projection_dim = 9216*4)
> > > | Neuron Activation | **A5000 Dense (Cublas) (ms)** | **Selective GEMM (ms)** | **Speedup** |   | **A100 Dense (Cublas) (ms)** | **Selective GEMM (ms)** | **Speedup** |   | **H100 Dense (Cublas) (ms)** | **Selective GEMM (ms)** | **Speedup** |
> > > | --- | ---: | ---: | ---: | - | ---: | ---: | ---: | - | ---: | ---: | ---: |
> > > | 0.10 | 0.99 | 0.11 | 8.86 | | 0.43 | 0.08 | 5.48 | | 0.24 | 0.05 | 5.26 |
> > > | 0.20 | 0.99 | 0.20 | 4.94 | | 0.43 | 0.10 | 4.37 | | 0.24 | 0.07 | 3.75 |
> > > | 0.40 | 0.99 | 0.40 | 2.50 | | 0.43 | 0.20 | 2.16 | | 0.24 | 0.10 | 2.45 |
> > > | 0.50 | 0.99 | 0.53 | 1.87 | | 0.43 | 0.22 | 1.91 | | 0.24 | 0.12 | 2.05 |
> > > | 0.70 | 0.99 | 0.70 | 1.43 | | 0.43 | 0.30 | 1.42 | | 0.24 | 0.17 | 1.45 |
> > > | 0.90 | 0.99 | 0.95 | 1.04 | | 0.43 | 0.39 | 1.09 | | 0.24 | 0.21 | 1.18 |
> > > | 1.00 | 0.99 | 1.00 | 1.00 | | 0.43 | 0.42 | 1.01 | | 0.24 | 0.25 | 0.98 |
> > >
> > > This confirms that both custom kernels in Polar Sparsity are efficient and generalize well to different modern GPUs.
> > >
> > > ## Final Remarks
> > >
> > > We hope these detailed responses fully address your remaining concerns. Your feedback has been invaluable in helping us strengthen the paper. We remain confident in the contributions of Polar Sparsity and would be grateful if you would consider our updated evaluation and clarifications.

---

### Comment · Area_Chair_HSb6 · 2025-08-03
**Reminder: Discussion Phase (July 31 – Aug 6)**

Hi everyone,

This is a reminder that the discussion phase is between July 31 – Aug 6.

Please read the author responses, especially where you are mentioned, and post your reply as soon as possible. This helps ensure there's time for meaningful back-and-forth.

Thanks for your engagement!

AC

---

### Note · Authors · 2025-08-13

We are deeply grateful to the AC and reviewers for an engaged discussion that measurably strengthened the paper. Throughout the discussion, we have demonstrated our commitment to addressing every concern with extensive new empirical evidence.

In direct response to the reviews, we have:

1. **Expanded Evaluation on Harder Tasks**: We added new, robust results on challenging benchmarks (MMLU-PRO, GSM8K, LongBench) and initiated ultra-long context evaluation (InfiniteBench). Our validation across **5 diverse language model families** confirmed our method’s robustness and effectiveness, showing an average accuracy degradation of just **~1%** and no catastrophic failures in long-form generation.

2. **Demonstrated Consistent, High-Impact Speedups**: Provided new benchmarks for our ready-to-use, open-source sparse kernels across consumer (A5000) and datacenter (A100, H100) GPUs. This confirms consistent, significant performance gains, culminating in up to **2.2x end-to-end speedup** in batched decoding and validating the real-world applicability of our work.

3. **Theoretical Sparsity to Wall-Clock Speedups**: Our work is the first to demonstrate significant, wall-clock speedups from contextual attention head sparsity in batched inference.  As our new theoretical roofline analysis and multi-GPU benchmarks show, this is a non-trivial systems achievement beyond the reach of modern compilers. Our kernels' unique **batch-invariant property** provides the essential building block for the next frontier: dynamic, **query-sensitive routing** to achieve potentially lossless sparse inference.

The dialogues with Reviewers sVp2, cRa2, and nrZF have reinforced the technical soundness of our approach, while our new long-context results for Reviewer bAJa confirm its scalability. We are confident Polar Sparsity provides a practical and foundational contribution for high-throughput LLM serving. We are already running the remaining InfiniteBench tasks and will include the complete evaluation in the final manuscript.

Thank you for reconsidering our work. We believe the revised paper, refined by this collaborative review process, will be a valuable addition to the NeurIPS 2025.

---

### Decision · Program_Chairs · 2025-09-17

**Decision:**

Accept (poster)

**Comment:**

The paper proposes Polar Sparsity, a batch-friendly framework that scales contextual sparsity to large-batch LLM inference, achieving up to 2.2× throughput gains across multiple models and hardware with minimal accuracy loss. While remaining concerns persist (e.g., downstream performance on reasoning and math tasks), the proposed method is well validated and delivers promising inference speedups, which is also integrable with existing solutions. Overall, this is a practical work that leverages the beneficial conflicts of contextual sparsity between Attention and MLP. I also suggest the authors to incorporate the valuable results from the rebuttal period into the revised manuscript.